# Using Machine Learning to Model Uncertainty for Water-Vapor Atmospheric Motion Vectors

Joaquim V. Teixeira[1], Hai Nguyen[1], Derek J. Posselt[1], Hui Su[1], Longtao Wu[1]

[1]Jet Propulsion Laboratory, California Institute of Technology

**Abstract.** Wind-tracking algorithms produce Atmospheric Motion Vectors (AMVs) by tracking clouds or water vapor across spatial-temporal fields. Thorough error characterization of wind-tracking algorithms is critical in properly assimilating AMVs into weather forecast models and climate reanalysis datasets. Uncertainty modelling should yield estimates of two key quantities of interest: bias, the systematic difference between a measurement and the true value, and standard error, a measure of variability of the measurement. The current process of specification of the errors in inverse modelling is often cursory and commonly consists of a mixture of model fidelity, expert knowledge, and need for expediency. The method presented in this paper supplements existing approaches to error specification by providing an error-characterization module that is purely data-driven. Our proposed error-characterization method combines the flexibility of machine learning (random forest) with the robust error estimates of unsupervised parametric clustering (using a Gaussian Mixture Model). Traditional techniques for uncertainty modeling through machine learning have focused on characterizing bias, but often struggle when estimating standard error. In contrast, model-based approaches such as k-means or Gaussian mixture modelling can provide reasonable estimates of both bias and standard error, but they are often limited in complexity due to reliance on linear or Gaussian assumptions. In this paper, a methodology is developed and applied to characterize error in tracked-wind using a high-resolution global model simulation, and it is shown to provide accurate and useful error features of the tracked wind.

## 1. Introduction

Reliable estimates of global winds are critical to science and application areas, including global chemical transport modeling and numerical weather prediction. One source of wind measurements consists of feature-tracking based Atmospheric Motion Vectors (AMVs), produced by tracking time sequences of satellite-based measurements of clouds or spatially distributed water vapor fields (Mueller et al., 2017; Posselt et al., 2019). The importance of global measurements of 3-dimensional winds was highlighted as an urgent need in the NASA Weather Research Community Workshop Report (Zeng et al., 2016) and was identified as a priority in the 2007 National Academy of Sciences Earth Science and Applications from Space (ESAS 2007) Decadal Survey and again in ESAS 2017. For instance, wind is used in the study of global $CO_2$ transport (Kawa et al., 2004), numerical weather prediction (NWP; Cassola and Burlando, 2012), as inputs into weather and climate reanalysis studies (Swail and Cox, 2000), and for estimating current and future wind-power outputs (Staffell and Pfenninger, 2016).

Thorough error characterization of wind-track algorithms is critical in properly assimilating AMVs into forecast models. Prior literature has explored the impact of 'poor' error-characterization in Bayesian-based approaches to remote sensing applications. Nguyen et al. (2019) proved analytically that when the input bias is incorrect in Bayesian

methods (specifically, optimal estimation retrievals), then the posterior estimates would also be biased. Moreover,
they proved that when the input standard error is 'correct' (that is, it is as close to the unknown truth as possible), then
the resulting Bayesian estimate is 'efficient'; that is, it has the smallest error among all possible choices of prior
standard error. Additionally, multiple active and passive technologies are being developed to measure 3D winds, such
as Doppler wind lidar (DWL), radar, and infrared/microwave sensors that derive AMVs using feature-tracking of
consecutive images. Therefore, an accurate and robust methodology for modeling uncertainty will allow for more
accurate assessments of mission impacts, and the eventual propagation of data uncertainties for these instruments.
Velden and Bedka (2009) and Salonen et al. (2015) have shown that height assignment contributes a large component
of uncertainty in AMVs tracked from cloud movement and from sequences of infrared satellite radiance images.
However, with AMVs obtained from water vapor profiling instruments (e.g., infrared and microwave sounders),
height assignment error cannot be directly assessed purely through analysis of the AMV extraction algorithm. Height
assignment is instead an uncertainty in the water vapor profile itself. Unfortunately, without the quantified
uncertainties on the water vapor profile necessary to pursue such a study, that is well beyond the scope of this paper.
As such, this study will focus on errors in the AMV estimates at a given height. Previous work has demonstrated
several different approaches for characterizing AMV vector error. One common approach is to employ quality
indicator thresholds, as described by Holmund et al (2001), which compare changes in AMV estimates between
sequential timesteps and neighboring pixels, as well as differences from model predictions, to produce a quality
indicator to which a discrete uncertainty is assigned. The Expected Error approach, developed by Le Marshal et al.
(2004), builds a statistical model using linear regression against AMV-radiosonde values to estimate the statistical
characteristics of AMV observation error.
In this study, we outline a data-driven approach for building an AMV uncertainty model using observing system
simulation experiment (OSSE) data. We build on the work by Posselt et al. (2019) in which a water vapor feature-
tracking AMV algorithm was applied to a high-resolution numerical simulation, thus providing a global set of AMV
estimates which can be compared to the reference winds produced by the simulation. In this case, a synthetic "true"
state is available with which AMVs can be compared and errors are quantified, and it is shown that errors in AMV
estimates are state dependent. Our approach will use a conjunction of machine learning (random forest) and
unsupervised parametric clustering (Gaussian mixture models) to build a model for the uncertainty structures found
by Posselt et al. (2019). The realism and robustness of the resulting uncertainty estimates depend on the realism and
representativeness of the reference dataset. This work builds upon the work of Bormann et al. (2014) and Hernandez-
Carrascal and Bormann (2014), who showed that wind tracking could be divided into distinct geophysical regimes by
clustering based on cloud conditions. This study supplements that approach with the addition of machine learning,
which, compared with traditional linear modeling approaches, should allow the model to capture more complex non-
linear processes in the error function.
Traditional techniques for modeling uncertainty through machine learning have focused on characterizing bias but
often struggle when estimating standard error. By pairing a random forest algorithm with unsupervised parametric
clustering, we propose a data-driven, cluster-based approach for quantifying both bias and standard error from
experimental data. According to the theory developed by Nguyen et al. (2019), these improved error characterizations
should then lead to improved error characteristics (e.g., lower bias, more accurate uncertainties) in subsequent analyses
such as flux inversion or data assimilation.
This paper does not purport that the specific algorithm detailed here should supplant error characterization approaches
for all AMVs; indeed, most commonly assimilated AMVs are based on tracking cloud features, not water vapor
profiles. In addition, this algorithm is trained and developed for a specific set of AMVs extracted from a water vapor
field associated with a particular range of flow features. As such, application of our algorithm to modeled or observed
AMVs will be most appropriate in situations with similar dynamics to our training set. However, we intend in this
paper to demonstrate that the methodology is successful in characterizing errors for this set of water vapor AMVs and
suggest that this approach— that is, capturing state-dependent uncertainties in feature-tracking algorithms through a
combination of clustering and random forest— could be implemented in other feature-tracking AMV extraction
methods and situations.
The rest of the paper is organized as follows: In Section 2, we give an overview of the simulation which provides the
training data for our machine learning approach. We then motivate and define the specific uncertainties this study
aims to characterize. In Section 3, we describe the error characterization approach with the specifics of our error
characterization model, including both the implementation of and motivations for employing the random forest and
Gaussian mixture model. In Section 4, we provide a validation of our methods, attempting to assess the bias of our
predictions. In Section 5, we discuss the implications of our error characterization approach, both on AMV estimation
and data assimilation more broadly.
**2. Experimental Set-up**
**2.1 Simulation and Feature-Tracking Algorithm**
We trained our model on the simulated data used by Posselt et al. (2019), which applied an AMV algorithm to outputs
from the NASA Goddard Space Flight Center (GSFC) Global Modeling and Assimilation Office (GMAO) GEOS-5
Nature Run (G5NR; Putman et al. 2014). The Nature Run is a global dataset with ~7 km horizontal grid spacing that
includes, among other quantities, three-dimensional fields of wind, water vapor concentration, clouds, and
temperature. Note that throughout the text we will use the term 'Nature Run wind' to refer to reference winds in the
simulation dataset used to train the uncertainty model. The AMV algorithm is applied on four pressure levels (300hPa,
500hPa, 700hPa, and 850hPa) at 6-hourly intervals, using three consecutive global water vapor fields spaced one hour
apart, and for a 60-day period from 07/01/2006 to 08/30/2006. The water-vapor fields from GEOS5 were input to a
local-area pattern matching algorithm that approximates wind speed and direction from movement of the matched
patterns. The algorithm searches a pre-set number of nearby pixels to minimize the sum-of-absolute-differences
between aggregated water vapor values across the pixels. Posselt et al. (2019) describes the sensitivity of the tracking
algorithm and the dependency of the tracked winds on atmospheric states in detail. The coordinates of the data are on
a 5758 x 2879 x 240 spatio-temporal grid for the longitude, latitude, and time dimension, respectively.
It is important to note that the AMV algorithm tracks water vapor on fixed pressure levels. In practice, these would be
provided by satellite measurements, whereas in this paper we use simulated water vapor from the GEOS-5 Nature
Run. In this simulation height assignment of the AMVs is assumed to be perfectly known. This assumption is far from
guaranteed in real world applications but, as previously discussed, its implications are not pursued in this paper. As
such, we focus solely on observational AMV error and not on height assignment error. We note that in practice, one
approach to understanding the behavior and accuracy of the wind-tracking algorithm is to apply it to modeled data
(e.g., Posselt et al., 2019). Our approach seeks to complement this approach by providing a machine-
learning/clustering hybrid approach that can further divide comparison domains into 'regimes' which may provide
further insights into the behavior of the errors and/or feedback into the wind-tracking algorithm.
A snapshot of the dataset at 700hPa is given in Figure 1, where we display the water vapor from Nature Run (top left
panel), the wind speed from Nature Run (top right panel), the tracked wind from the AMV-tracking algorithm (bottom
right panel), and the difference between the Nature Run and tracked wind (bottom left panel). Note that the wind-
tracking algorithm tends to have trouble in region where the Nature Run water vapor content is close to zero. It is clear
that while the wind-tracking algorithm tends to perform well in most regions (we can classify these regions as areas
where the algorithm is skilled), in some regions the algorithm is unable to reliably make a reasonable estimate of the
wind speed (unskilled). We will examine these skilled and unskilled regimes (and their corresponding contributing
factors) in section 3.
**2.2 Importance of Uncertainty Representation in Data Assimilation**
Proper error characterization for any measurement, including AMVs, is important in data assimilation. Data
assimilation often uses a regularized matrix inverse method based on Bayes' theorem, which, when all probability
distributions in Bayes' relationship are assumed to be Gaussian, reduces to minimizing a least-squares (quadratic) cost
function Eq (1):

$$\mathbf{J} = (\mathbf{x} - \mathbf{x_b})\mathbf{B}^{-1}(\mathbf{x} - \mathbf{x_b}) + \left((\hat{\mathbf{y}} - \mathbf{a}) - \mathbf{H}[\mathbf{x}]\right)^{\mathrm{T}}\mathbf{R}^{-1}\left((\hat{\mathbf{y}} - \mathbf{a}) - \mathbf{H}[\mathbf{x}]\right) \qquad (1)$$

where $\mathbf{x}$ represents the analysis value, $\mathbf{x_b}$ represents the background field (first guess), $\mathbf{B}$ represents the background
error covariance, $\mathbf{y}$ represents the observation, and $\mathbf{H}$ represents the forward operator that translates model space into
observation space. This translation may consist of spatial and/or temporal interpolation if $\mathbf{x}$ and $\mathbf{y}$ are the same variable
(e.g., if the observation of temperature comes from a radiosonde), or may be far more complicated (e.g., a radiative
transfer model in the case of satellite observations). $\mathbf{R}$ represents the observation error covariance, and $\mathbf{a}$ represents
the accuracy, or bias, in the observations. The right hand side of Eq. (1) can be interpreted as a sum of the contribution
of information from the data ($\mathbf{y} - \mathbf{H}[\mathbf{x}]$ - $\mathbf{a}$) and the contribution from the prior ($\mathbf{x} - \mathbf{x_b}$), which are weighted by their
respective covariance matrices. In our analysis, the AMVs obtained from the wind-tracking algorithm is used as 'data'
in subsequent analysis. That is, the tracked wind data $\hat{\mathbf{y}}$ is a biased and noisy estimator of the true wind $\mathbf{y}$, and might
be assumed to follow the model Eq. (2):

$$\hat{y} = y + \epsilon \tag{2}$$

where $\epsilon$ is an error term, commonly assumed to be Gaussian with mean $\mathbf{a}$ and covariance matrix $\mathbf{R}$ (i.e., $\epsilon \sim N(\mathbf{a}, \mathbf{R})$),
which are the same two terms that appear in Equation (1). As such, for data assimilation to function, it is essential to
correctly specify the bias vector $\mathbf{a}$ and the standard error matrix $\mathbf{R}$. Incorrect characterizations of either of these
components could have adverse consequences on the resulting data assimilation analyses with respect to bias and/or
the standard error (Nguyen et al., 2019).

## 3 Methodology

### 3.1 Generalized Error Characterization Model

An overview of our approach is outlined in Figure 2. Given a set of training predictors X, training responses $\hat{Y}$, and
simulated true response Y, our approach begins with two independent steps. In one step, a Gaussian mixture model is
trained on the set of X, $\hat{Y}$, and Y. This clustering algorithm identifies geophysical regimes where the nonlinear
relationships between the three variables differ. In the other step, a random forest is used to model Y based on X and
$\hat{Y}$. This step produces an estimate of the true response (we call this $\tilde{Y}$) using only the training predictors and response.
We then employ the Gaussian mixture model to estimate the clusters which the set of X, $\hat{Y}$, and $\tilde{Y}$ pertain to.
Subsequently, we compute the error characteristics of each cluster of X, $\hat{Y}$, and $\tilde{Y}$ in the training dataset. Thereafter,
given a new point consisting solely of X and $\hat{Y}$, we can assign it to a specific cluster and ascribe to it a set of error
characteristics.
In this paper, we are primarily interested in the distribution of a retrieved quantity versus the truth. That is, given a
retrieved value $\hat{Y}_i$, we are interested in the first and second moments (i.e., $E(\hat{Y}_i - Y)$ and $var(\hat{Y}_i - Y)$), respectively.
We note that there is a large body of existing work on uncertainty modeling in the machine learning literature (e.g.,
Coulston et al., 2016; Tripathy et al., 2018; Tran et al., 2019; Kwon et al., 2020), although these approaches primarily
define the uncertainty of a prediction as $var(\hat{Y}_i)$, or quantify how sensitive that prediction is to tiny changes in the
models/inputs. Our approach, on the other hand, characterizes the error as $var(\hat{Y}_i - Y)$, which describes how accurate
a prediction is relative to the *true value*. For this reason, our methodology is more stringent in that it requires
knowledge of the true field (which comes naturally within OSSE framework) or some proxies such as independent
validation data or reanalysis data. In return, the error estimates from our methodology fit naturally within the data
assimilation framework (that is, it constitutes the parameter R in Eq. (1)).
What follows in this paper is an implementation of the error characterization model obtained for a subsample of the
GEOS-5 Nature Run at a fixed height of 700hPa. In particular, we trained the error characterization on a random
subsample from the first 1.5 months of the Nature Run, and show the results obtained when applying it to a test
subsample drawn from the subsequent 0.5 months of the Nature Run.

## 3.2 Error Regime

When examining the relationship between AMVs and Nature Run winds in Figure 3, it is clear that there are two
distinct 'error-regimes' present in the dataset. The majority of AMV estimates can be categorized as 'skilled', wherein
their estimate lies clearly along a one-to-one line with the Nature Run wind. However, there is also clearly an
'unskilled' regime, for which the AMV estimate is very close to zero when there are actually moderate or large Nature
Run wind values present. Our goal is to provide unique error characterizations for each error regime, because the error
dynamics are different within each regime. Furthermore, when we analyze this error and its relationship to water
vapor, we see that 'unskilled' regime correlates highly with areas of low water vapor in Figure 4. This matches the
error patterns discussed in Posselt et al. (2019). We note that the division between skilled and unskilled regimes does
not need to be binary. For instance, in some regions the wind-tracking algorithm might be unbiased with high-
correlation with the true winds, and in other regions the algorithm might still be unbiased relative to the true winds,
but with higher errors. The second situation is clearly less skilled than the first, although it might still be considered
'skilled', and this separation of the wind-tracking estimates into various 'grades' of skill forms the basis of our error
model.

## 3.3 Gaussian Mixture Model

These distinct regimes present an opportunity to employ machine learning. Bormann et al. (2014) and Hernandez-
Carrascal and Bormann (2014) demonstrated that cluster (also called regime) analysis is a successful approach for
wind-tracking error characterization, and so we aim to train a clustering algorithm that will cluster a given  individual
AMV estimate to various 'grades' of skill. In particular, we use a clustering algorithm that can take advantage of the
underlying geophysical dynamics. To this end, we employ a Gaussian mixture model, an unsupervised clustering
algorithm based on estimating a training set as a mixture of multiple Gaussian distributions.  A mathematical overview
follows:
1. Define each location containing Nature Run winds, water vapor, and AMV estimates as a random variable

$x_i$

2. Define $\theta$ as the population that consists of all $x_i$ in the training dataset
3. Model the distribution of the population $P(\theta)$ as:

$$P(\theta) = \sum_{j}^{K} \pi_j N(\mu_j,\ \Sigma_j) \tag{3}$$

Where $N(\mu_j,\ \Sigma_j)$ is the normal distribution with mean $\mu_j$ and covariance $\Sigma_j$ of the *j*-th cluster,

K is the number of clusters, and $\pi_j$ is the mixture proportion.
4.   Determine $\pi_j, \mu_j, \Sigma_j$ for K clusters using the Expectation–Maximization Algorithm
5.   From 3. and 4., estimate the probability of a given $x_i$ belonging to the j-th cluster as $P(x_i \in k_j) = p_{ij}$
6.   Assign point $x_i$ to the cluster with the maximum probability $p_{ij}$
The mixture model clustering is based on the R package 'Mclust' developed by Fraley et al. (2012), which builds upon
the theoretical work of Fraley and Raftery (2002) for model-based clustering and density estimation. The process uses
an Expectation-Maximization algorithm to cluster the dataset, estimating a variable number of distinct multivariate
Gaussian distributions from a sample dataset. Training the Gaussian mixture model on this dataset provides a
clustering function which outputs a unique cluster for any data point with the same number of variables.
In one dimension, a Gaussian mixture model looks like the distributions depicted in Figure 5: instead of modeling a
population as a single distribution (Gaussian or otherwise), the GMM algorithm fits multiple Gaussian distributions
to a population. One key aspect of this algorithm is the capability of assigning a new point to the most likely
distribution. For example, in the 1-D figure, a normalized AMV estimate with a value of 10 would be more likely to
originate from the broad cluster '2' than the narrow cluster '4'. In this case, we model the population as a Gaussian
mixture model in five-dimensional space, which consists of two Nature Run wind vector components (u and v), two
AMV estimates of these wind components (û and v̂), and the simulated water vapor values, all of which have been
standardized to have mean 0 and standard deviation of 1. Each cluster has a 5-dimensional mean vector for the center
and a 5x5 covariance matrix defining their multivariate Gaussian shape. The estimation of a covariance matrix allows
for the characterization of the relationships between the different dimensions within each cluster, and as such the
gaussian mixture model approach provides greater potential for understanding the geophysical basis of error regimes
than other unsupervised clustering approaches.
We note that the choice of inputs to the clustering methodology is limited, and that a more successful clustering may
be achieved by including additional meteorological or geographic information. However, the intention of this paper
is to study the ability of a purely data-driven approach, where no additional information or assumptions are passed to
the machine learning model outside of the inputs and outputs to the AMV algorithm itself. Posselt et al. (2019) showed
that state dependent uncertainties are a major source of error in water vapor AMVs; introducing further information
may cloud our ability to discern these specific uncertainties. While scaling this methodology to other applications may
incentivize tailoring to specific conditions, this paper aims to demonstrate that modifications are encouraged for
improvement, but not necessary for success.
Having trained the Gaussian mixture model on the 1.5 month training dataset, we applied the clustering algorithm to
a testing dataset sampled from the subsequent 0.5 months of the nature run. By re-analyzing the AMV estimate in
relation to the Nature Run winds within each cluster (**Error! Reference source not found.**), we find that the clustering
approach successfully separates the AMV estimates according to their 'skillfulness'. Essentially, we repeat Figure 3
but divide the AMV estimates by cluster. We see that, for example, clusters 4, 5, and 7 clearly represent cases in which
the feature-tracking algorithm provides an accurate estimate of the Nature Run winds, with very low variance around
the one-to-one line (i.e., low estimation errors). Clusters 1, 2, 3, and 9 are somewhat noisier than the low-variance
clusters, with error characteristics similar to those of the entirety of the dataset. That is, they are considered less skilled,
but their estimates still lie on a one-to-one line with respect to the true wind. Clusters 6 and 8, on the other hand, are
clearly unskilled in different ways. Cluster 6 is a noisy regime, which captures much of the more extreme differences
between the AMV estimates and the Nature Run winds. Cluster 8, on the other hand, represents the low AMV estimate,
high Nature Run wind regime. This cluster is returning AMVs with values of zero where the Nature Run wind is
clearly non-zero because of the very low water vapor present. We further see the stratification of the regimes when
analyzing the absolute AMV error in relation to the water vapor content (Figure 7). We see that clusters that have
similar behaviors in the error pattern (such as 1, 2, and 3) represent different regimes of water vapor content.
We specified 9 individual clusters due to a combination of quantitative and qualitative reasons. Quantitatively, the
'Mclust' package uses the Bayesian Information Criterion (BIC), a model selection criterion based on the likelihood
function which attempts to penalize overfitting, to select the optimal number of clusters given an input range. Using
an input range of one through nine, the BIC indicated the highest number of clusters would be optimal. More
importantly, however, the 9 clusters can be physically distinguished and interpreted. Plots of the geophysical variables
in the testing set associated with each of the clusters are shown in Figures 8-11. Specifically, Figure 8 plots the
distribution of water vapor for each cluster, while Figure 9 plots the mean wind magnitude in each direction by cluster.
Figure 10 plots the correlation matrix for each cluster and Figure 11 show the geographic distribution of each cluster.
In looking at these in combination, we see discernable and discrete clusters with unique characteristics. For example,
cluster 1 captures the very dry, high-wind regime in the southern hemisphere visible in Figure 2. Cluster 7
encompasses the tropics, while cluster 3 captures mid-latitude storm systems. Clusters 6, 8, and 9 are all characterized
by a much worse performance of the AMV tracking algorithm, exhibited both in Figure 7 and in Figure 8 but all
encompass different geographic and geophysical regimes. We see that the clustering algorithm succeeds in capturing
physically interpretable clusters without having any knowledge of the underlying physical dynamics. We note that in
other applications, the optimal number of clusters will change and the researcher will need to explore various choices
of this parameter in their modeling, although this tuning process should be greatly simplified by the inclusion of an
information criterion (e.g., BIC) in the GMM algorithm.

**3.5 Random Forest**

The clustering algorithm requires the Nature Run wind vector component values (u and v) in order to classify the
AMV error. When applying the algorithm in practice to tracked AMV wind from real observations, the true winds are
unknown. To represent the fact that we will not know the true winds in practice, we develop a proxy for the Nature
Run winds using only the AMV estimates and the simulated water vapor itself. This is an instance in which the
application of machine learning is desirable, since machine learning excels at learning high-dimensional non-linear
relationships from large training datasets. In this case, we specifically use random forest to create an algorithm which
predicts the Nature Run wind values as a function of the tracked wind values and water vapor.
Random forest is a machine learning regression algorithm which, as detailed by Breiman (2001), employs an ensemble
of decision trees to model a nonlinear relationship between a response and a set of predictors from a training dataset.
Here, we chose random forest specifically because it possesses certain robustness properties that are more appropriate
for our applications than other machine learning methods. For instance, random forest will not predict values that are
outside the minimum and maximum range of the input dataset, whereas other methods such as neural networks can
exceed the training range, sometimes considerably so. Random forest, due to the sampling procedure employed during
training, also tends to be robust to overtraining in addition to requiring fewer tuning parameters compared with
methods such as neural networks.
We trained a random forest with 50 trees on a separate set of tracked winds and water vapor values to predict Nature
Run winds using the 'randomForest' package in the R programming language. While the random forest estimate as a
whole does not perform much better than the AMV values in estimating the Nature Run wind (2.89 RMSE for random
forest vs 2.91 RMSE for AMVs), as shown in Figure 12, it does not display the same discrete regimentation as the
AMV estimates in Figure 3. As such, the random forest estimates can act as a proxy for Nature Run wind values in
our clustering algorithm — they remove the regimentation which is a critical distinction between the AMV estimates
and the Nature Run wind values.
**3.6 Finalized Error Characterization Model**
The foundation of the error characterization approach is to combine the random forest and clustering algorithm. We
apply the Gaussian mixture model, as trained on the Nature Run winds (in addition to the AMVs and water vapor), to
each point of water vapor, AMV estimate, and associated random forest estimate. This produces a set of clusters
which, when implemented, require no direct knowledge of the actual Nature Run state (Figure 13).
Naturally, the clustering algorithm performs better when applied to the dataset with the Nature Run winds, as
opposed to winds generated from the random forest algorithm. The former is created with direct knowledge of the
Nature Run winds, and any approximation will lead to increased uncertainties. In practice, the performance of the
cluster analysis can be improved by enhancing the performance of the random forest itself. As with any machine
learning algorithm, the random forest contains hyperparameters that can be optimized for specific applications. In
addition, performance could be improved by including additional predictor variables. Our intent is not to use the
random forest as a wind tracking algorithm; rather, the random forest is presented in this paper as a proof of concept.
Nonetheless, we see in Figure 13 and Figure 14 that the error characterization still discretizes the testing data set into
meaningful error regimes. The algorithm manages to separate the AMV estimates into appropriate error clusters. Once
again, clusters 6 and 8 manage to capture unskilled regimes, and cluster 7, and to a lesser extent clusters 4 and 5,
remain skillful. By taking the mean and standard deviation of the difference between AMV estimates and Nature Run
winds in each cluster, we develop error characteristics for each cluster (Figure 15); these quantities are precisely the
bias and uncertainty that we require for the cost function J in Eq (1). We see that the unskilled clusters have very high
standard errors and they correspond roughly to the areas of unskilled regimes in Figure 3. Similarly, skilled clusters
5, 4 and 7 have standard errors below that of the entire dataset. Since each cluster now has associated error
characteristics (e.g., bias and standard deviation), it is then straightforward to assign the bias and uncertainty for any
new tracked wind observation by computing which regime it is likely to belong to.

## 3.7 Experimental Set up

In this section we will describe our experimental setup for training our model on the GEOS-5 Nature Run data and
testing its performance on a withheld dataset. We divide the dataset into two parts: a training set consisting of the first
1.5 months of the GEOS-5 Nature Run, and a testing set consisting of the last 0.5 month of the Nature Run. Our
training/testing procedure for the simulation data and tracked wind is as follows:

1. Divide the simulation data and tracked wind into two sets: training set of 1,000,000 points from the first 1.5 months of the Nature Run and a testing set of 1,000,000 points from the final 0.5 months of the Nature Run.
2. We train a Gaussian Mixture Model on a normalized random sample of observations from the training dataset of Nature Run winds (u and v direction), tracked winds (u and v direction), and water vapor with n=9 clusters.
3. We train two separate random forests on a different random sample of 750,000 observations from the training dataset. We use tracked wind (u and v direction) and water vapor to model, separately, Nature Run winds in both the u and v directions.
4. We apply the random forests to the dataset used for the Gaussian Mixture Model. This provides a random forest estimate for each point, which is used as a substitute for Nature Run wind values in the next step.
5. We predict the Gaussian mixture component assignment for each point of water vapor, tracked winds, and random forest estimate using the GMM parameters estimated in Step 2.
6. We compute the mean and standard deviation of the difference between the tracked winds and the Nature Run winds, per direction, for each Gaussian mixture model cluster assignment. This provides a set of error characteristics that are specific to each cluster.
7. We can apply the random forest, and then the cluster estimation, to any set of water vapor and tracked AMV estimates. Thusly, any set of tracked AMV estimates and water vapor can be mapped to a specific cluster, and therefore its associated error characteristics.

## 4 Results and Validation

In this section, we compare our clustering method against a simple alternative, and we quantitatively demonstrate
improvements that result from our error characterization. Recall that in Section 3, we divided the wind-tracking
outputs into 9 regimes, which range from very skilled to unskilled. For the $i$-th regime, we can quantify the predicted
uncertainty estimate as a gaussian distribution with mean $m_i$ and standard deviation $\sigma_i$, which has a well-defined
cumulative distribution function which we denote as $F_i$. To test the performance of our uncertainty forecast, we divide
the dataset described in Section 2 into a training dataset (first 1.5 month) and a testing dataset (last 0.5 month). Having
trained our model using the training dataset, we apply the methodology to the testing dataset, and we compare the
performance of the predicted probability distributions against the actual wind error (tracked winds - Nature Run
winds). This is a type of probabilistic forecast assessment, and we assess the quality of the prediction using a scoring
rule called continuous ranked probability score (CRPS), which is defined as a function of a cumulative distribution
function F and an observation x as follows:
$$\mathrm{CRPS}(\mathbf{F}, \mathbf{x}) = \int_{-\infty}^{\infty} (\mathbf{F}(\mathbf{x}) - \mathbb{1}(\mathbf{y} - \mathbf{x}))^2 \, \mathbf{dy} \qquad\qquad (\mathbf{4})$$
Where $\mathbb{1}( )$ is the Heaviside step function and denotes a step function along the real line that is equal to 1 if the argument
is positive or zero, and it is equal zero if the argument is negative (Gneiting and Katzfuss, 2014) . The continuous rank
probability score here is strictly proper, which means that the function $\mathrm{CRPS}(\mathbf{F}, \mathbf{x})$ attains the minimum if the data x
is drawn from the same probability distribution as the one implied by F. That is, if the data x is drawn from the
probability distribution given by F, then $\mathrm{CRPS}(F, x) < \mathrm{CRPS}(G, x)$ for all $G \neq F$.
The alternative error characterization method that we test against is a simple marginal mean and marginal standard
deviation of the entire tracked subtract Nature Run  wind dataset. This is essentially equivalent to an error
characterization scheme that utilizes one regime, where m and $\sigma$ are given as the marginal mean and the marginal
standard deviation of the residuals (i.e., tracked wind minus Nature Run  winds). Here, we use a negatively oriented
version of the CRPS (i.e., Eq.(4) without the minus sign), which implies that lower is better. A histogram evaluating
the performance of our methodology against the naive error characterization method is given in Figure 16.
The relative behavior of the CRPS is consistent between u and v winds. The CRPS tends to have to wider distribution
when applied to the regime-based error characterization. Compared to the alternative error characterization scheme,
our methodology produces a cluster of highly accurate predictions (low CRPS scores), in addition to some cluster of
very uninformative predictions (high CRPS scores). These clusters correspond to the highly skilled cluster (e.g.,
Cluster 3) and the unskilled clusters (Cluster 6 and 8), respectively. Overall, the mean of the CRPS is lower for our
methodology than it is for the alternative method, indicating that as a whole our method produces a more accurate
probabilistic forecast.
Thus far we have shown that our method produces more accurate error-characterization than an alternative method
based on marginal means and variance. Now, we assess whether our methodology provides valid probabilistic
prediction; that is, we test whether the uncertainty estimates provided are consistent with the empirical distribution of
the validation data. To assess this, we construct a metric in which we normalize the difference between the Nature
Run wind and the tracked wind by the predicted variance. That is, for the $i$-th observation, we compute the normalized
values for $u_i$ and $v_i$ using the following equations:

$$z_{u,i} = \frac{u_i - \hat{u}_i}{\sigma_{u,i}}$$


$$z_{v,i} = \frac{v_i - \hat{v}_i}{\sigma_{v,i}}$$

(5)

Where $u_i$ is the *i*-th Nature Run u wind from the Nature Run data, $\hat{u}_i$ is the tracked-wind, and $\sigma_{u,i}$ is the error as
assessed by our model (recall that it is a function of the regime index to which $\hat{u}_i$ has been assigned). The values for
the v-wind are defined similarly. The residuals in Eq (5) can be considered as a variant of the z-score, and it is
straightforward to see that if our error estimates are valid (i.e., accurate), then the normalized residuals in Eq. (5)
should have a standard deviation of 1. If our uncertainty estimates $\sigma_{u,i}$ and $\sigma_{v,i}$ are too large, then the standard deviation
of $z_{u,i}$ and $z_{v,i}$ should be less than 1; similarly, if our uncertainty estimates are too small, then the standard deviation
of $z_{u,i}$ and $z_{v,i}$ should be larger than 1. In *Figure 17,* we display the histogram of the normalized residuals $z_u$ and $z_v$.
It is clear that for both types of wind, the standard deviation of $z_{u,i}$ and $z_{v,i}$ are 1.003 and 1.009, respectively, indicating
that our error characterization model is highly accurate when forecasting uncertainties.
A further validation of our methods encompasses an analysis of the statistical significance of the uncertainty in our
model. To this end, we constructed confidence intervals for the bias and standard deviation within each regime using
the bootstrap (Efron and Tibshirani, 1993). The procedure of our bootstrap is as follows

1.  Subset the data to retain only observations with regime index j. Let's assume that we have $N_j$ observation
within this data subset
2.  Sample *with replacement* $N_j$ observations from this subset. This forms a bootstrap sample
3.  From 2., compute an estimate of the bias and standard deviation.
4.  Repeat step 2-3 for 1000 times, giving us 1000 estimates of the bias and 1000 estimates of the standard
deviation within regime j.
5.  Compute 95% confidence intervals from the 1000 estimates of bias and standard deviation from 4.

The results for the confidence intervals (in graphical form in Figure 18. We note that the figure indicates that for
many of the biases, they can be considered unbiased since their confidence interval includes 0 (e.g., regimes 2-8 for
u-wind). However, the plot also clearly indicates that two regimes are statistically different from 0 (regime 1 and 9).
We also note that for the standard deviation maps, the CI's indicate that they are fairly stable (small narrow range)
and that most of the regimes have statistically different standard deviation (denoted here visually as CI's that do not
overlap one another). We also note that u and v wind direction tend to have very similar patterns, indicating that our
regime classification is persistent across u and v. To summarize, the CI plot above indicate that the differences in
standard deviation between different regimes are highly statistically significant (as evidenced by the small
confidence intervals and their spacing). For the biases, 3 of the regimes are statistically significantly different from
the rest (i.e., regimes 1, 6, and 9), while the rest are likely relatively unbiased (i.e., bias = 0 ).

**5 Conclusion and Discussion**

Error characterization is an important component of data validation and scientific analysis. For wind-tracking algorithms, whose outputs (tracked u and v) are often used as observations in data assimilation analyses, it is necessary to accurately characterize the bias and standard error (e.g., see Section 2.2). Nguyen et al. (2019) illustrated that incorrect specification of these uncertainties ($\mathbf{a}$ and $\mathbf{R}$ in Eq. (1)) can adversely affect the assimilation results – mischaracterization of bias will systematically offset a tracked wind, while an erroneous standard error could incorrectly weigh the cost function.

In this paper we demonstrate the application of a machine learning uncertainty modeling framework to AMVs derived from water vapor profiles intended to mimic hyper-spectral sounder retrievals. The methodology, based on a combination of gaussian mixture model clustering and random forest, identified distinct geophysical regimes and provided uncertainties specific to each regime. This was achieved in a purely data-driven framework; nothing was known to the model except the specific inputs and outputs of the AMV algorithm, deducing the relationship between regime and uncertainty from the underlying multivariate distribution of water vapor, Nature Run wind, and tracked wind. Our algorithm does require one major tuning parameter in the number of clusters for the GMM algorithm, although the search for the 'optimal' number of clusters can be aided by the inclusion of an information criterion (e.g., the BIC) in the GMM model. This implementation is not intended as a 'ready-to-go' algorithm for general use. Instead, we lay the foundation of an uncertainty modelling approach which we plan to implement at a larger scale in subsequent work Nonetheless this bare bones implementation is sufficient to produce improved error estimates of state-dependent uncertainties as detailed in Posselt et al. (2019).

We introduce this framework in an environment that is limited and well-behaved, but which nonetheless we believe provides insight into how such an approach would perform at a larger scale. Of course, there are issues when moving from the controlled environment of the simulation study to large scale applications. We understand these to be: (1) the existence of uncertainty on the tracked humidity values, and (2) the ability of the training dataset to adequately capture both the range of conditions of water vapor and wind speed, and their inherent relationship.

The simulation used for introducing this framework was a 'perfect-observation' environment; that is, the water vapor was assumed to be perfectly known to the wind tracking algorithm. In real world scenarios, this is obviously not the case. However, we believe that this is mitigated by two factors. Firstly, Posselt et al (2019) also conducted a study where measurement noise was added to the water vapor measurement. This did not show to have an effect on the uncertainty in the AMV estimate, except where there was the presence of strong vertical wind shear, a situation which can be identified a larger scale application. Secondly, given quantified uncertainties on the water vapor retrievals themselves (the scope of which is decidedly outside the work of this paper), these could be assimilated into the uncertainty modelling framework in a straightforward manner by adding them as a prediction variable in both the regime classification and emulator. This would allow for the model to itself ascertain the relationship

between water vapor uncertainty and AMV estimate uncertainty, without breaking the foundational aspect of being
data-driven.
The reliability of the training dataset is the fundamental assumption of any machine learning approach. To reiterate,
we present a methodology which aims to characterize the uncertainty in the difference between a measurement $\hat{X}$
and its true target $X$ (that is, var ($\hat{X} - X$)). As such, we require some proxy for the truth in the development of our
model (call this $X^*$). To expand further, we are modelling the relationship between $\hat{X}$ and $X$ as a function of water
vapor $Y$, with $f(Y) = \hat{X}$ and $g(Y) = X$, where $f$ represents the AMV algorithm and $g$ the 'true' relationship
between wind speed and water vapor. Thus, we additionally require a proxy function $g^*$, which is the relationship
implied by the training data output of water vapor and reference winds. In the implementation presented in this
paper, $g^*$ is represented by the underlying physical models that model the motion of water vapor and windspeed in
the GEOS-5 Nature Run.
The fidelity of our framework relies upon the assumption $X^* \sim X$ and $g^* \sim g$. In the simulation study, $X^*$ is the first
1.5 months of a nature run simulation, which is used as a proxy for an $X$ which consists of the last .5 months of a
nature run simulation. We have given the algorithm a training dataset with what we believe is a plausible range of
conditions which could occur in $X$. To the extent that errors may be seasonally and regionally dependent, it will be
more effective to train the error estimation algorithm on data that is expected to represent the specific flow regimes
and water vapor features valid for a particular forecast or assimilation period. A range of model data encompassing
enough seasonal variability should be a reasonable proxy for the possible range of true $X$. This would significantly
increase the computational demands of training the model (~1 day on a single processor, per pressure level to train
the current implementation of the algorithm and an average of 3 days per pressure level, on a non-optimized cluster
network to run the AMV extraction on the nature run), although such concerns could be mitigated by strategic
subsampling approaches.
On the other hand, in this implementation $g^*$ is a perfectly known representation of $g$, which is the GEOS-5 model
that runs the simulation. This is where the simulation approach might create the largest source of uncertainty and
unreliability in the model. The true process g can only ever be approximated, and different attempts to do so will
involve different tradeoffs when implementing this framework. Users could, for example, use high quality validation
data such as matchups with radiosondes. In theory, this provides the best possible approximation of the true process
$g$, but could involve a sparsity of data such that the range of, $X^*$ supplied is too narrow for a useful model (indeed,
the data might be so sparse as to— from a pure machine learning aspect— reduce the overall fidelity of the model
itself). On the other hand, model or reanalysis data can provide dense and diverse training datasets, but rely on the
assumption that the underlying physical models in those simulations are an adequate representation of the true
process. At the core of atmospheric models such as GOES-5 are the laws of fluid dynamics and thermodynamics. In
this context, water vapor is advected by the mean wind and as such the wind and water vapor are intrinsically related
in these models. This has been the case since the first atmospheric weather prediction models have been developed.
There are of course uncertainties associated with the discretization of the fluid dynamics equations, and sometimes
also with parameterizations depending on the physical constraints. But these uncertainties are likely small for the
water vapor structures that are selected for the wind tracking algorithm.
In both these cases, the model could likely be improved by the inclusion of additional variables in the clustering
algorithm. These could include a variety of parameters to address different potential problem areas in the model. As
mentioned previously, including quantified values of uncertainty in water vapor estimates would algorithmically
link the uncertainty in the humidity retrieval with the uncertainty in the AMV tracking. Similarly, including
parameters that correlate with geophysical phenomena where the AMV algorithm is known to perform poorly (such
as a marker for vertical wind shear or frontal features) would enable domain knowledge to inform the clustering
algorithm and emulator. Finally, it is likely that the several parameters used in formulating both the Quality
Indicator (Holmlund et al. 1998) and Expected Error (Le Marshall et al. 2004) approaches would be informative in
enhancing the algorithm. One critical aspect for users to consider is that these variables must be continuous
parameterizations, rather than discrete markers (which are often used in quality control); discrete variables cannot be
easily incorporated into a Gaussian mixture model, or indeed most clustering algorithms. Furthermore, we would
recommend that users implement parameters that are readily available at the same measurement location and time as
the AMV estimate itself. Part of the motivation for the purely state dependent approach in this framework is ease of
implementation; colocation and interpolation could add further uncertainty to the model.
We note that in real applications, using a proxy X* instead of the true X will result in our algorithm estimating the
variability $\text{var}(\hat{X} - X^*)$ instead of $\text{var}(\hat{X} - X)$. Therefore, the degree to which $\text{var}(\hat{X} - X^*)$ approximates $\text{var}(\hat{X} -$
$X)$ relies on the accuracy on the proxy data relative to the true uncertainty. Ultimately, implementing this
methodology at scale requires confidence in the training dataset employed by the user. As with most machine
learning approaches, a thorough understanding of the relative strengths and weaknesses of the training dataset is the
most critical consideration for users. This means not only ensuring that the training data is variable and diverse
enough to encapsulate the entirety of the true domain, but possessing some understanding of how and where
portions of the training dataset might be less representative of reality. There are a few practical ways in which users
could attempt to address this issue. Given adequate resources and time, users could train the uncertainty model under
various training datasets. While this would not necessarily give a greater understanding of the training data's
relationship with the truth, the differences between the produced models would provide some quantification of the
effect of the training data on the estimated uncertainties. Similarly, if users have some quantified understanding of
areas wherein the training dataset might be less useful (e.g., collocation errors), they could leverage this to inform
the uncertainty model. In this case, it is likely such decisions would manifest themselves in the final uncertainty
product. Nonetheless, as much as users should try to mitigate the potential for problems, there is always an
underlying leap of faith that they have chosen a training dataset that adequately represents the truth in their
application. Like any modeling approach, this methodology relies on a set of assumptions; this is one such
assumption. This is why domain knowledge is critical in developing a similar uncertainty model. Thoughtful and
careful implementations by users, keeping in mind the prescriptions and concepts detailed above, should mitigate the
training data dependent uncertainty.
Future users would also be wise to consider improvements in the random forest step of the framework. The
capability of this implementation in discerning accurate error regimes degrades substantially with the introduction of
the random forest wind estimates. This work focused on the ability to capture regime dependent error, and as such
the random forest was not studied in depth. An improved emulator would certainly increase the accuracy of the
uncertainty estimates produced by this framework. There are a wide variety of ways to improve the emulator;
ultimately, and even more so than the regime classification, these will be specific to the AMV extraction algorithm
being used. Certainly, many of the additional variables suggested above could be useful towards improving the
random forest. Users could also investigate replacing the random forest altogether with a different emulator, such as
a neural net or a gaussian process. Indeed, at its most general, our methodology consists of two parts: an emulator
and a clustering algorithm. In this implementation, random forest and Gaussian mixture modelling are the
approaches; in theory, these two steps could be accomplished using other algorithms belonging to the appropriate
class.
Thorough domain knowledge, both of the AMV extraction algorithm and the context in which it will be applied, is
critical in developing methods to improve it. As discussed previously, the bare bones implementation of our
methodology in this paper is intended as a structural presentation of the conceptual framework, not necessarily a
finalized model. However, it is also the case that the investigation by Posselt et. al (2019) showed that the variables
used in this implementation of the model are those most strongly related with AMV uncertainty in this particular
application.  The state-dependent errors identified by Posselt et al. (2019) are also expected to apply to other water
vapor AMVs. This is because, in general, AMV algorithms have difficulty tracking fields with very small gradients,
and will produce systematic errors in situations for which isolines in the tracked field (e.g., contours of constant water
vapor mixing ratio) lie parallel to the flow. To the extent that our algorithm represents a general class of errors, the
results may be applicable to other geophysical scenarios and other AMV tracking methodologies. As mentioned in the
introduction, robust estimates of uncertainty are important for data assimilation, and we expect that our methodology
could be used to provide more accurate uncertainties for AMVs used in data assimilation for weather forecasting and
reanalysis.

**Author Contribution**

Teixeira conceived of the idea with inputs from Nguyen. Teixeira performed the computation. Wu provided the
experimental datasets along with data curation expertise. Posselt and Su provided subject matter expertise. All authors
discussed the results. Teixeira wrote the initial manuscript and updated the draft with inputs from co-authors.
**Competing Interest:** The Authors declare no conflict of interest.
**Funding Acknowledgment**: The research was carried out at the Jet Propulsion Laboratory, California Institute of
Technology, under a contract with the National Aeronautics and Space Administration (80NM0018D0004). © 2020.
California Institute of Technology. Government sponsorship acknowledged

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

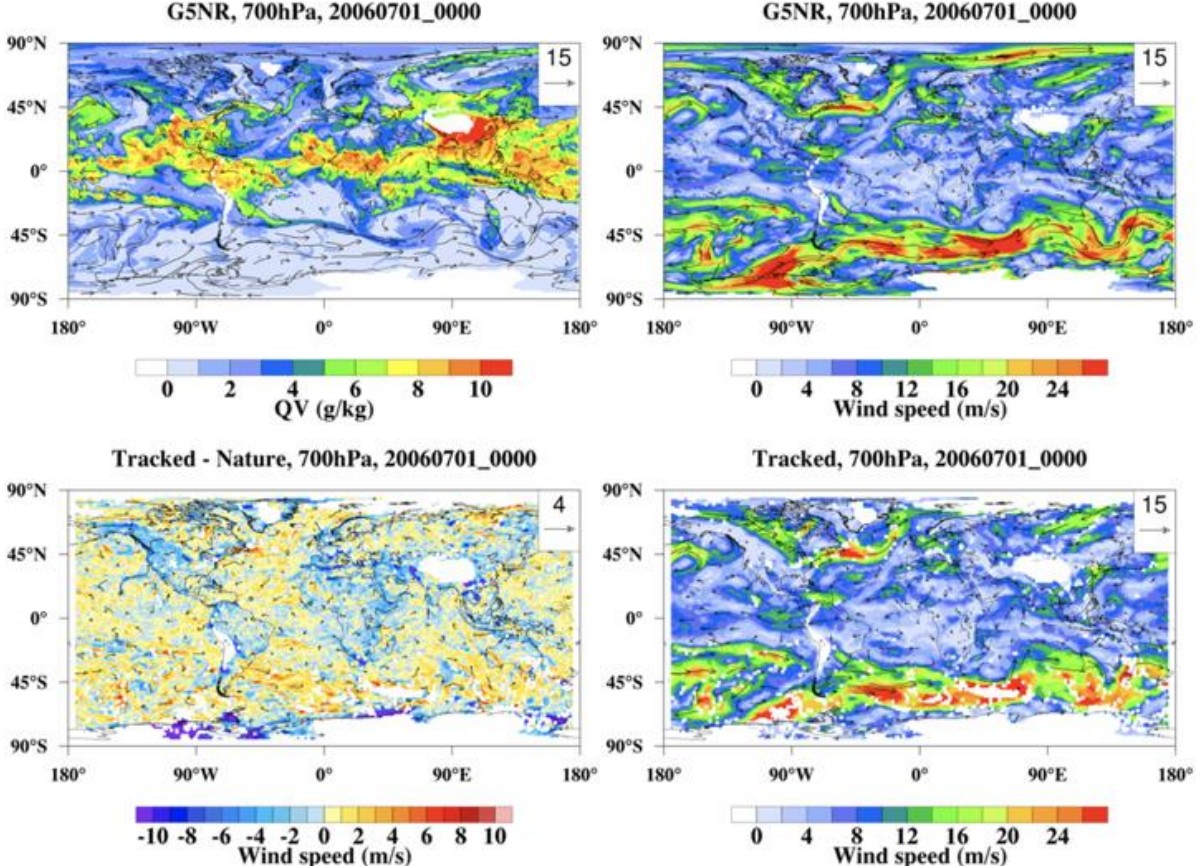

**Figure 1: Map of Nature Run at one timestep at 700hPa (A): Water Vapor (B): Nature Run Wind Speed (C): Difference between Nature Run Wind Speed and AMV Estimate (D): AMV Estimate.**

## 1. Training

**Training Dataset**
$X$: Water Vapor
$\hat{Y}$: AMV Estimates
$Y$: Simulated True Winds

**2 Gaussian Mixture Model**
Clustering Algorithm *GMM* models ($X$, $\hat{Y}$, $Y$) into clusters $I$

**1 Random Forest**
Random Forest function *RF* generates $\tilde{Y} \sim Y$ for all pairs of $X$ and $\hat{Y}$

**Error Regimes**
Let $\hat{I} = GMM$ ($X$, $\hat{Y}$, $\tilde{Y}$)
Every set of ($X$, $\hat{Y}$, $Y$) has an associated cluster $\hat{I}$

**3 Error Characteristics**
For every cluster $\hat{I}$:
$\mu_{\hat{I}} = mean\ (\hat{Y}_{\hat{I}} - Y_{\hat{I}})$
$\sigma_{\hat{I}} = \sqrt{var\ (\hat{Y}_{\hat{I}} - Y_{\hat{I}})}$

**Model Output**
**1** RF  **2** GMM  **3** *Bias $\mu_{\hat{I}}$*
*Standard Error $\sigma_{\hat{I}}$*

## 2. Implementation

**New Output**
$X$: Water Vapor
$\hat{Y}$: AMV Estimates

**1 Random Forest**
$\tilde{Y} = RF(X, \hat{Y})$

**2 Clustering**
$\hat{I} = GMM$ ($X$, $\hat{Y}$, $\tilde{Y}$)

**3 Error Values**
$\hat{Y}$ has **bias $\mu_{\hat{I}}$** and **standard error $\sigma_{\hat{I}}$**

 **Figure 2: Diagram of Training Approach and Diagram of Implementation steps.**

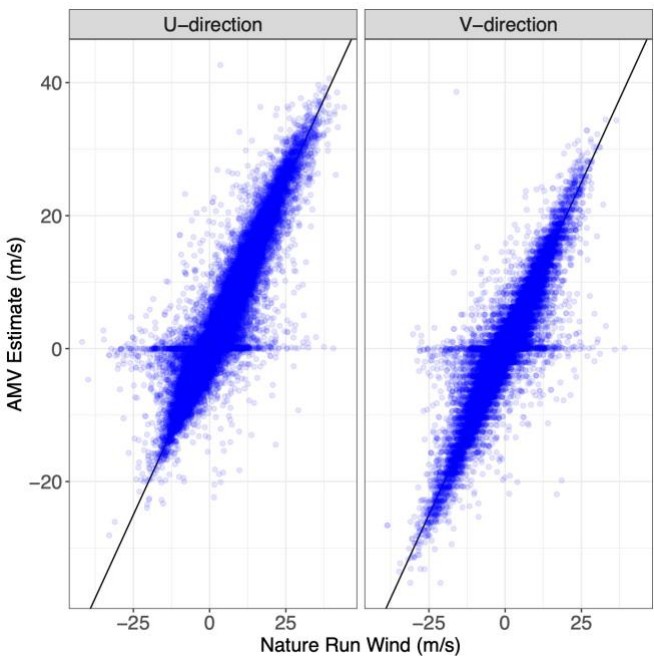


Figure 3: Scatter plot of the simulated Nature Run wind vs AMV estimates for u and v wind in the training dataset.


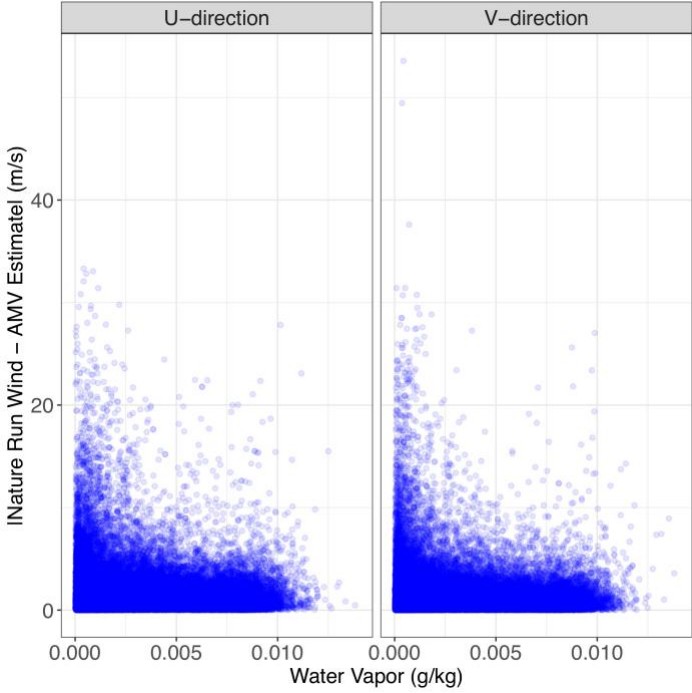


Figure 4: Simulated water vapor vs the absolute value of the difference between Nature Run and tracked winds in the training dataset.


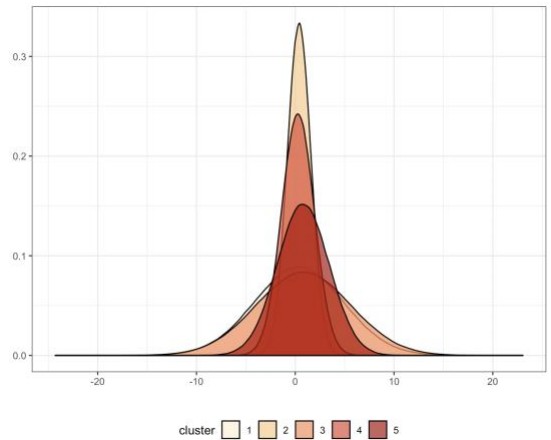


**Figure 5: Example of Gaussian Mixture Model in one dimension. Density Figures for the U-Direction AMV**


**Estimate dimension of fitted Gaussian mixture.**



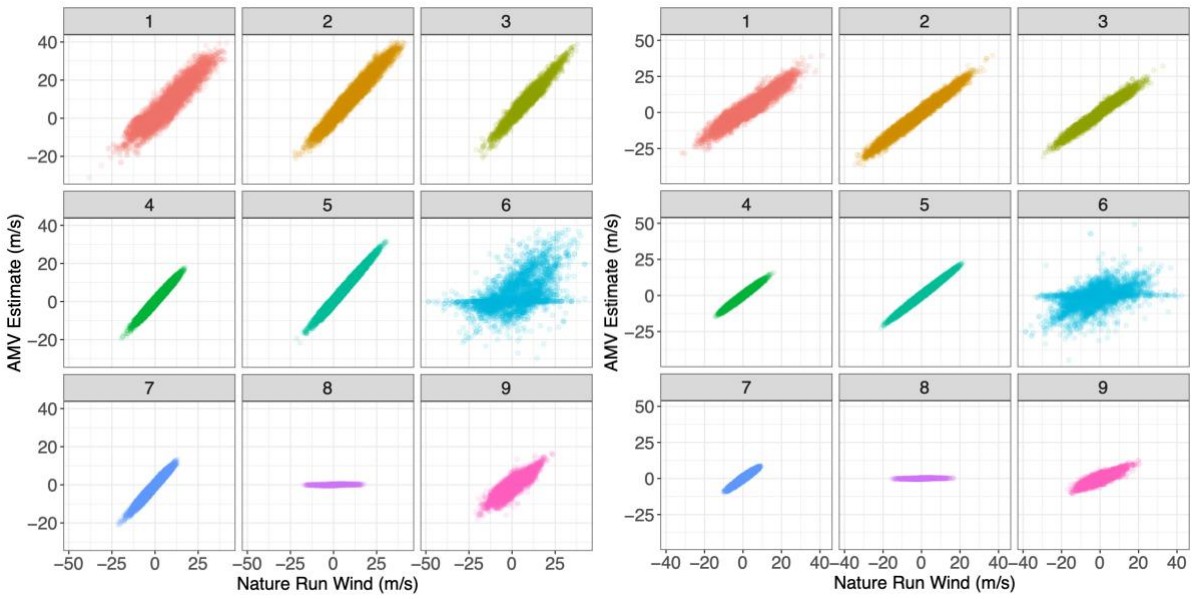


**Figure 6: Scatterplot of simulated Nature Run wind vs AMV Estimates, each sub-panel corresponding to the**


**specific Gaussian mixture component to which each point in the testing set has been assigned. (A): U-**


**Direction Wind (B): V-Direction Wind.**


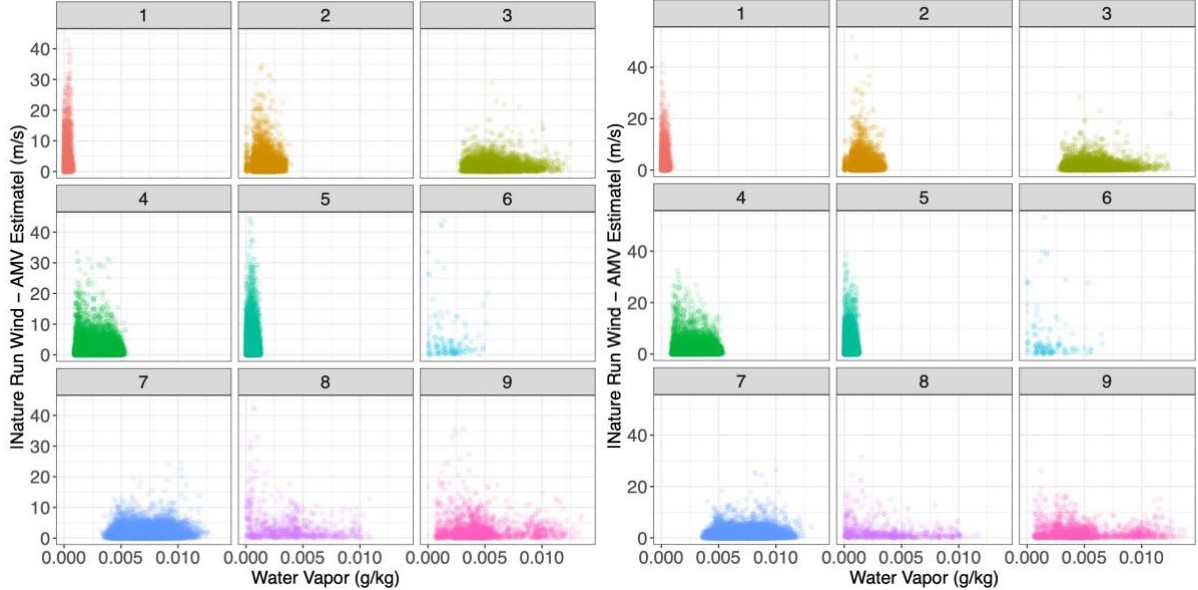


**Figure 7: Scatterplot of Water Vapor vs Absolute Tracked Wind Error, each sub-panel corresponding to the specific Gaussian mixture component to which each point in the testing set has been assigned. (A): U-Direction Wind (B): V-Direction Wind.**

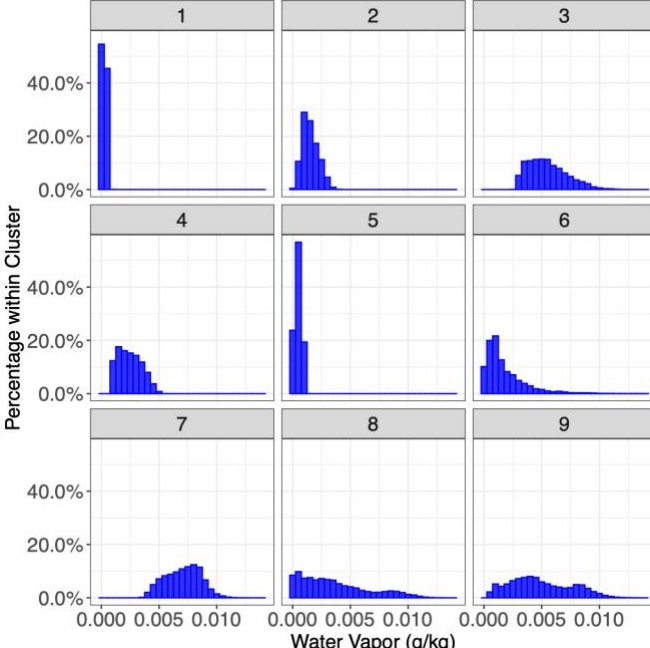

**Figure 8: Histogram of Nature Run water vapor for each cluster identified by the Gaussian mixture model, applied to the testing set. Each sub-panel represents the cluster each point was assigned to.**

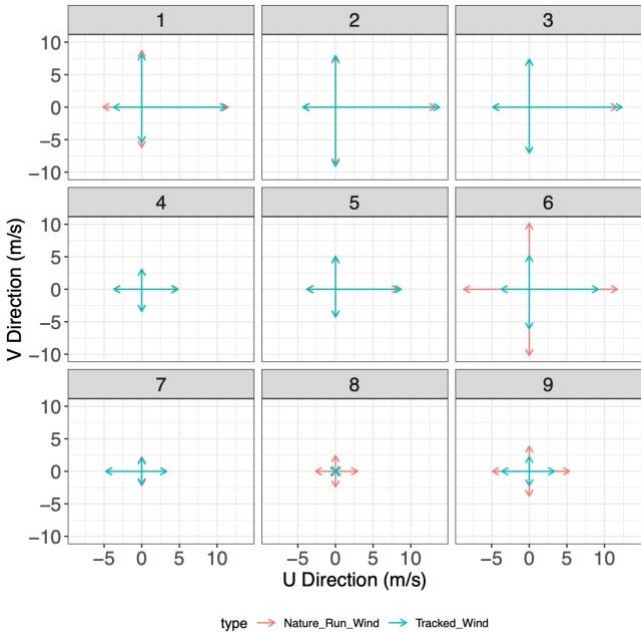

**Figure 9: Mean tracked winds and Nature Run winds, in each direction, for each cluster applied to the test**
**set. Each sub-panel represents the cluster each point was assigned to.**

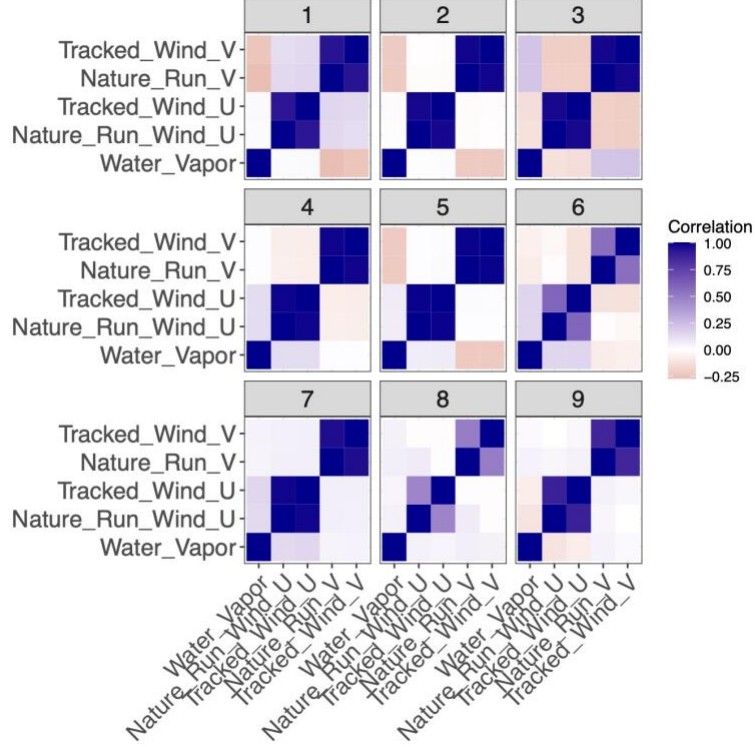

**Figure 10: Correlation matrix between each clustered element for each identified cluster in the original**
**training dataset. Each sub-panel refers to a specific cluster.**

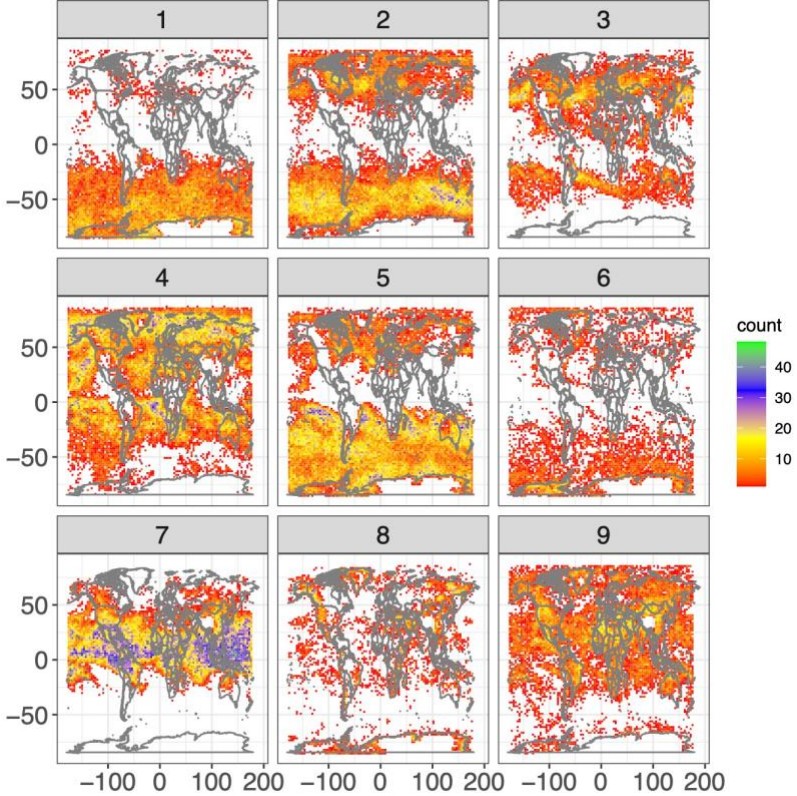

**Figure 11: Geographic distribution by cluster of AMV retrieval locations in the testing dataset. Each sub-**
**panel represents one cluster.**


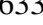


**Figure 12: Scatterplot of Nature Run wind estimate vs random forest produced estimate. (A): U Direction (B): V Direction**



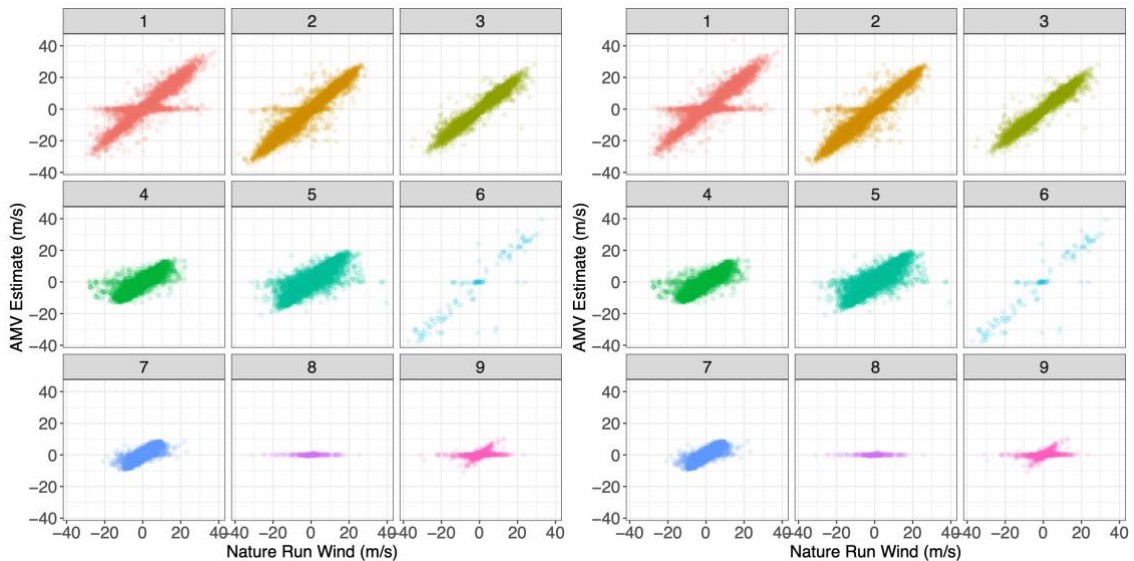

**Figure 13: Scatterplot of Nature Run wind vs AMV Estimates, each sub-panel corresponding to the specific Gaussian mixture component to which each point in the testing set has been assigned when the Nature Run wind value has been substituted by the random estimate. (A): U-Direction Wind (B): V-Direction Wind**

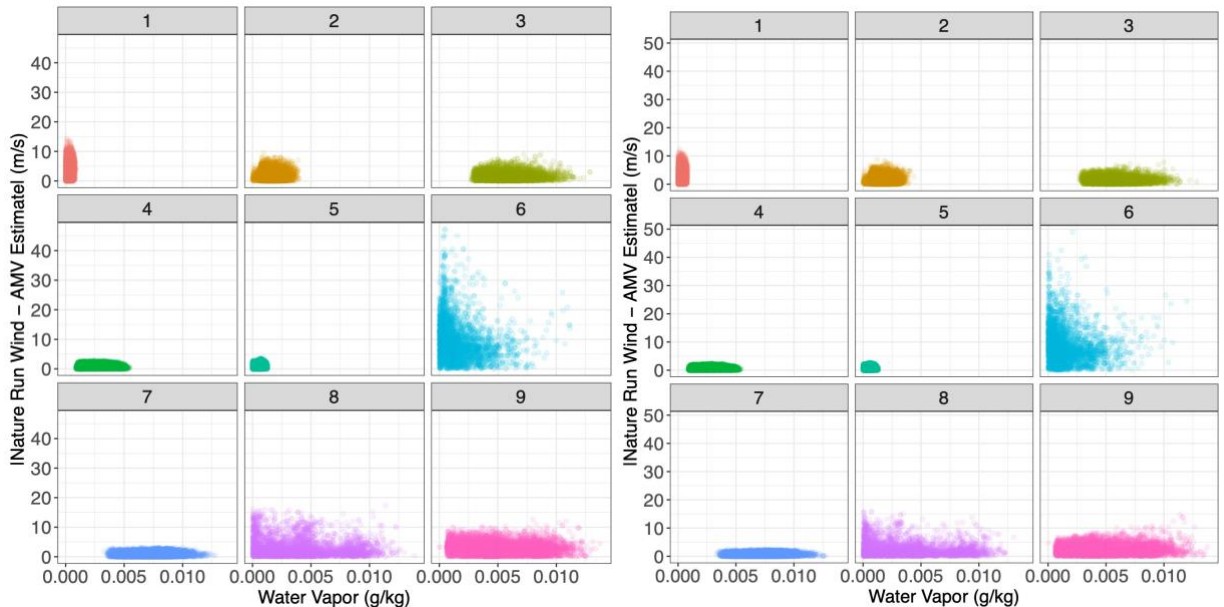


**Figure 14: Water Vapor vs Absolute Tracked Wind Error, each sub-panel corresponding to the specific Gaussian mixture component each point in the testing set has been assigned when the Nature Run wind value has been substituted by the random estimate. (A): U-Direction Wind (B): V-Direction Wind**

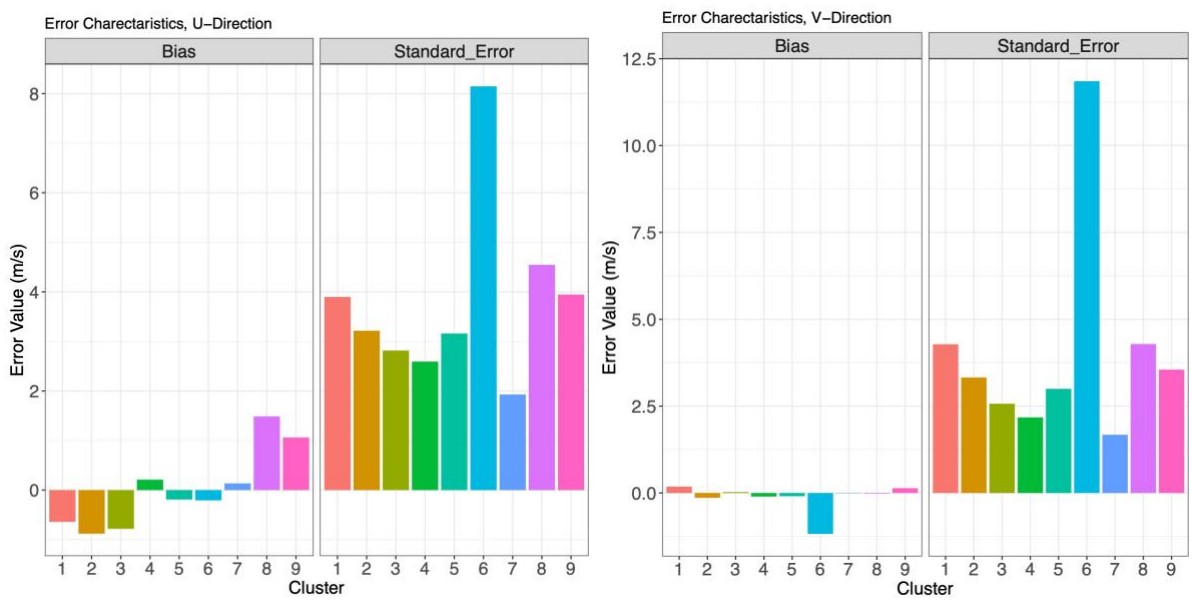


**Figure 15: (A): Bias (Left Panel) and Standard Error (Right Panel) for each Gaussian mixture cluster in**


**figure 6, U direction. (B): Same as (A) for V-direction**


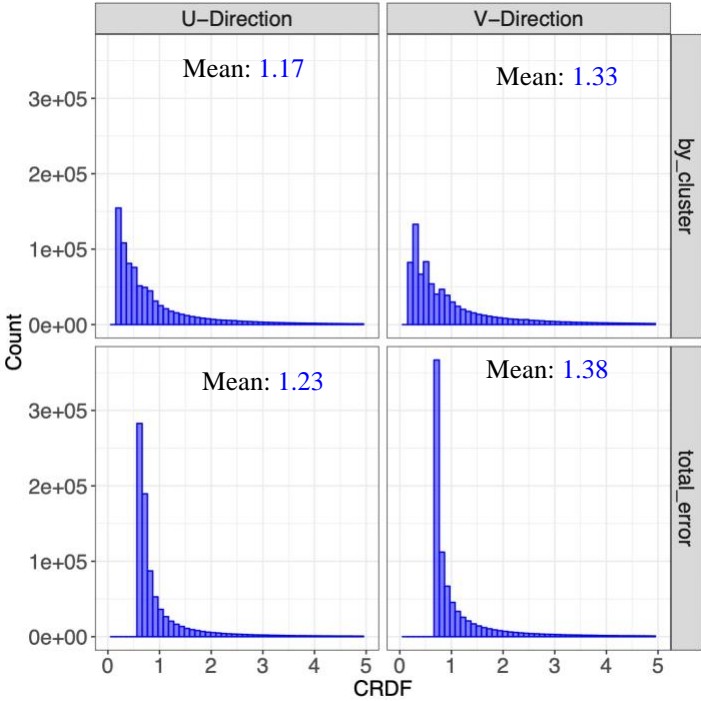


**Figure 16: CRSP applied to different error approaches. (A): Cluster Errors for U Winds (B): Total Errors**


**for U Winds (C): Cluster Errors for V Winds (D): Total Errors for V Winds.**



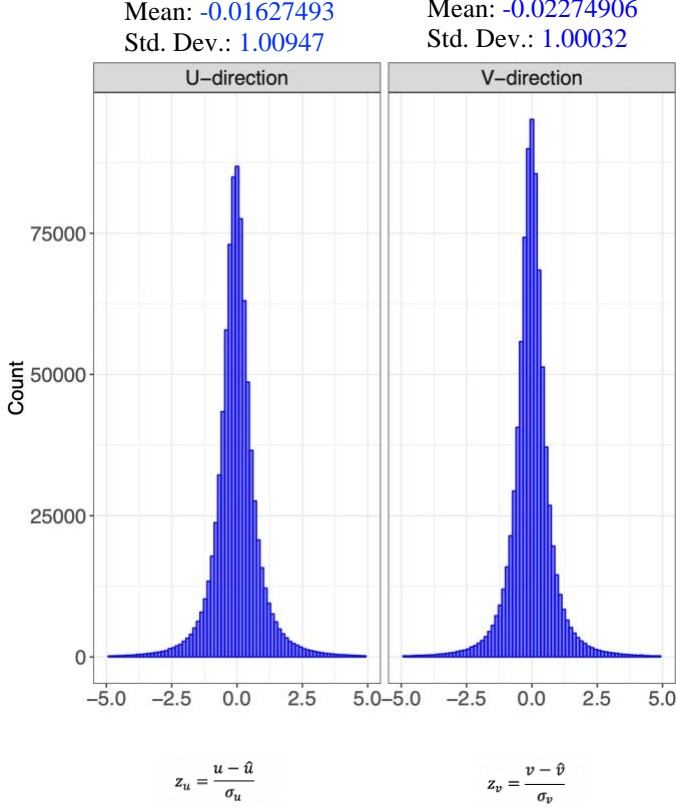



**Figure 17: U and V winds normalized using the error characteristics developed by our methodology.**












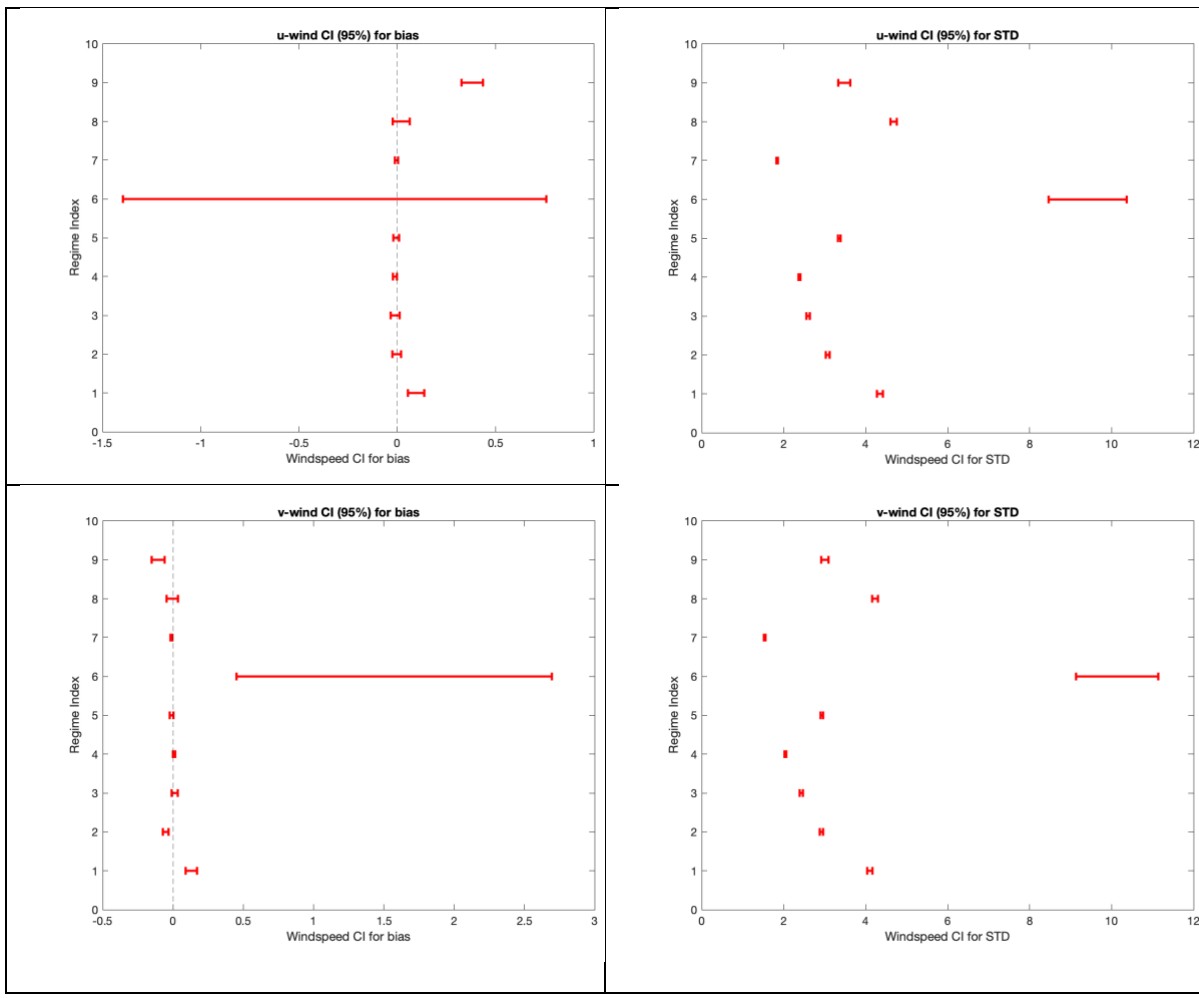

**Figure 18: Top rows (bias and std confidence intervals for u-wind), bottom rows (bias and std confidence**
**intervals for v-winds). The interval represent a 95% confidence interval.**



