# Peer review of "Using Machine Learning to Model Uncertainty for Water-Vapor Atmospheric Motion Vectors"

_Atmospheric Measurement Techniques, 2020_

## Referee Comment (RC1) · Anonymous Referee #1 · 6 May 2020

GENERAL COMMENTS

This paper presents a method of uncertainty characterization for Atmospheric Motion Vectors that is based on machine learning approach. The method has been applied to AMVs extracted from several water vapour layers of simulated data outputs from GEOS-5 Nature Run. The 2 months dataset used in the study has been split in two parts: training dataset (1.5 month) and testing dataset (0.5 month) for the validation. The results effectively show capabilities to filter skilled and unskilled regimes, and to produce some error estimates.

The paper is well written and well organized, presenting the method in section 3 and

the results and validation in section 4 before the concluding remarks.

I was personally disappointed by this paper that sounded very promising at the beginning. My main criticisms are:

- I have a problem with the present title. Reading it the first time I thought that paper was about improving error/quality of AMVs during the extraction process, and not during the assimilation process. From my understanding a title like: 'Use of Machines Learning to improve Uncertainty Quantification of Atmospheric Motion Vectors assimilated in NWP models', would certainly match better the real content of the paper and be less confusing

- Test presented in this paper is limited to water vapour AMVs extracted on specific layers. This potentially corresponds to extraction of 3D winds from hyperspectral sounders, as mentioned in the introduction. However, there is actually no evidence that the results can be generalised to the common AMVs extracted from clouds tracking in infrared or visible channels. If the method is limited to hyperspectral winds, this must be clearly specified in the text and probably also in the title of the paper, and not let the reader supposed that it works for all types of AMVs. If the method is not limited to hyperspectral AMVs authors have to present results also with common cloud motion winds extracted from satellite imagery. I understand from the text that another paper is upcoming (line 325), but there is no description or information that can actually let me assume that common AMVs have been used, and that the results are positive.

- The algorithm seems to be too dependent on the user's choice of the number of clusters, and the paper does not discuss the dependence of the algorithm on the chosen training dataset. It is also very unclear if the different clusters identified could refer to kind of physical or geographical AMVs properties, or if they are only blindly resulting out of the numerical tests. - Authors must clarify/discuss if the results may depend on the AMV extraction model used (Mueller 2017). It is not clear if the same clusters can be used for operational AMV extracted from other schemes too (NOAA, EUMETSAT,

JMA. . .Etc). If it is not the case I guess this study must be repeated individually for every different AMV extraction schemes and maybe after every releases of these codes, which should represent an important limitation for operational use in NWP models.

Although the authors promise the possibility to distinguish different Âń geophysical regimes Âż, the application ultimately presented by the paper comes down to discriminating the AMVs that are null because they are tracking the ground radiance, which is much too simple to showcase the real benefits of the algorithm.

Therefore I finally decided to reject this paper. However I encourage the authors to submit a new version, including new inputs based on the comments above and additional results applied to real AMV observations.

I also mention specific comments below. SPECIFIC COMMENTS

1) Everywhere I would change the denomination "true wind" to "G5NR wind" throughout the text. No matter the quality of any dataset relating to physical quantities, it does not deserve to be called "true".

2) Line 144 It would be good to recall that this Figure relates to the first 1.5 months of the dataset, in the caption of the Figure.

3) Lines 144-145 This is disappointing. Given the use of a powerful tool like GMM and the possibility of identifying "geophysical regimes" (line 132), I expected far more than just discriminating two groups, one being functional AMVs, and the other merely being the AMVs tracking the ground radiance, when the water vapour layer is too thin.

4) Line 270 This parts misses a "is" between "xi" and "the".

5) Section 4 The term Continuous Ranked Probability Score should be mentioned at least once before the formula at line 278. The two acronyms CPRS and CRPS are used in this section. Please correct.

6) Line 309 You are referring to Figure 13, and not Figure 12 as written.

7) Lines 329-330 I find your conclusion a little daring, knowing that you had to try different numbers of clusters before actually managing to discriminate the null AMVs.

---

## Referee Comment (RC2) · Anonymous Referee #2 · 25 Jun 2020

The paper summarises a study that uses a Gaussian Mixture Model to describe error characteristics of Motion Vectors derived from single-level model humidity fields. The Gaussian Mixture model is trained with the values of the "truth" available. To be able to apply this without the true wind field available, a random forest approach is applied to estimate the true wind from the derived Motion Vectors and an estimate of water vapour. The approach aims to provide situation-dependent estimates of typical errors in the Motion Vectors, and this is highly relevant for data assimilation applications. The manuscript is clearly written and mostly well-structured. While I'm not fully convinced about the practical applicability of the results for real-life observations, I feel the general methodology described may indeed offer novel approaches and hence merits publica-

tion, but the limitations and shortcomings need to be more critically assessed, as well as the physical motivation for some of the choices. Subject to addressing the points outlined below, I think this could be achieved in a major revision.

General points:

1. My main criticism of the study is that I am unsure about the practical applicability of the results. The study relies on the "truth" being available from a nature run to train the algorithm in the first place (e.g., to derive the clustering, to derive the random forest). It is unclear to me how this will be circumvented for real-life applications, without introducing other problems that may jeopardise the performance of the algorithm. I am not convinced that the algorithm could be applied "as is" on Motion Vectors derived from humidity fields retrieved from real sounding data, and indeed no attempt is presented in the paper to investigate this. The paper should discuss how it is envisaged that the algorithm can be applied to real-life situations and what the potential problem areas are.

2. In several areas the manuscripts appears to suggest that the method would be generally applicable, ie to other AMVs and possibly beyond (e.g., p3 L80 "... our methodology in principle could be used to quantify uncertainty in any measurements..."). I think this should be qualified. Subject to the point above, the algorithm may offer some value for AMVs derived from sounder retrievals; I suspect the value for the cloud-tracked AMVs is very limited - though these are currently the most widely used AMV datasets. There may be applicability beyond this, but the authors should explain more clearly how they expect the algorithm to be applied to "any measurement".

3. It would be useful if the authors took a critical look at the physical basis or motivation of their algorithm. The algorithm attempts to provide an uncertainty estimate for a derived wind vector with the derived wind vector and water vapour as the only inputs. I would expect other factors to play a considerable role, such as predictors describing the texture of the scene (to characterise the likely success of the tracking step), or

predictors that describe more the meteorological conditions (to characterise how likely humidity features are passive tracers). Spatial consistency measures such as the ones typically used in the formulation of the Quality Indicator (Holmlund 1998) may also be relevant. The predictor choice used in the study appears ad-hoc to me, and it could almost certainly be improved.

Specific points:

1. Title: I find the title misleading, as the authors only address the uncertainty in the wind estimates, not the height assignment uncertainty, which is a leading contributor of uncertainty for the most commonly used AMVs. The use of "Atmospheric Motion Vectors" may also lead readers to believe they will read about cloudy-tracked winds, when the links to these in the manuscript are very weak. I suggest to be more specific in the title, maybe "Estimation of uncertainty in wind retrievals derived from tracking humidity structures using Machine Learning".

2. p2, L34: Nguyen et al (2019) is referred to quite extensively in the paper (here and elsewhere), but is listed as a comparatively inaccessible report from the National Institute for Applied Statistics Research Australia. A journal paper with a similar title has recently been published, and I wonder whether this could be referred to instead.

3. p2, L 44-45 "However, height assignment is not the dominant portion of the error...": This is a strong claim to make, and I think it needs to be backed up with a suitable reference. Retrievals from infrared or microwave sounders do not represent radiosonde-like profiles. For a given level in the retrieved profile, the averaging kernel will describe the characteristics in the vertical represented by the retrieval - and these are not Dirac-delta functions. Height characteristics of AMVs derived from such retrievals will hence be rather complex, and interpreting them subsequently as single-level winds may well be a considerable contribution to the error budget. I am not aware that this aspect has been thoroughly investigated in the literature yet. It should at least be mentioned in the present study.

4. p2, L51 "The Expected Error . . . to correct AMV observation error.": The EE aims to provide an estimate of the statistical characteristics of the observation error, but does not try to correct any errors in the AMVs. Please rephrase.

5. p3, L90/91: It would be useful to provide an idea of the spatial scales used in the tracking step, ie what is the typical size of the target used.

6. p 3, L 100/101, Fig. 1: The authors emphasise the poorer performance in drier regions. While it is a little harder to see, my impression is that there is also poorer performance near frontal features (e.g, positive biases East of South America or East of North America). Poorer performance around frontal regions seems physically plausible, as single-level humidity may not be a passive tracer in these regions. I think it would be worth commenting on this in the main text. This could also motivate a predictor other than water vapour in the scheme developed later.

7. p 5, L139-142: It is not quite clear to me whether the description of the training/testing dataset in this paragraph is effectively referring to the same datasets described later (p8 L248/249). I got the impression here that all data for the 1.5/0.5 months were used, but later it sounds as if the dataset was subsampled. I suggest making this clearer to avoid confusion.

8. p 6, L187-191: It would be good if the authors could motivate further how they chose 9 clusters in the Gaussian mixture model. The text sounds as if it was a subjective choice, but maybe there was an objective component as well? Given the very limited inputs to characterise the conditions, and the lack of clear distinctions between some clusters, the chosen number of clusters appears high.

9. p 7, L224/225 "Relative to . . . entire dataset.": I am unsure about what is meant here. I suggest rephrasing.

10. p 8, first paragraph: It looks to me as if the clustering algorithm performs significantly more poorly once the true wind value has been substituted. Contrary to what is

said in the text, clusters 4 and 5 shown in Fig. 9 appear relatively unskilful, certainly in comparison to the same clusters shown in Fig. 6. Also, it looks as if the population in clusters 6 and 8 (referred to as the "unskilled" regimes) is very low, and much lower than what was found in Fig. 6. It appears that the assignment into these clusters is very different to what was possible before. This may not be too surprising, as the previous assignment had the benefit of the truth being available, but the aspect is not addressed much in the text.

11. p8, second paragraph/Fig. 11: Are the differences in standard deviation or bias between the clusters statistically significant? Also, what is the relative population of each cluster? Judging by Fig. 9 and 10, the clusters with the most different standard deviation (clusters 6 and 8) appear to have relatively small populations, whereas the variation in standard deviation in the remaining clusters is smaller.

12. p 8, L248/249: The authors mention that they use a training set of 1,000,000 points, and a testing dataset of the same number of points. How have these been chosen within the available data? It looks as if many more points were available, at least for the training dataset. Also, the link to p 5 L139-142 was not quite clear to me.

13. p 9, formula 4 and elsewhere: Typo: CPRS should be CRPS.

14. p 9, L279-283: The "$\leq$" in L282 appears to be inconsistent with what is said about CRPS earlier in the paragraph.

15. Fig. 12 and 13: Are these showing results for the test dataset? I assume they do (based on what is said on p 5, L141/142), but I think it would be clearest if this information was provided in the caption (a similar comment could be made for Fig. 6-11).

16. p 10, L306-311: The authors point to the finding that the residuals normalised with the estimated error have a standard deviation close to 1. It's a useful cross-check, but I suspect this finding primarily reflects that the training and testing data has similar

standard deviations of AMVs vs true winds. I suspect it would have been obtained by assigning one constant observation error equal to the standard deviation of the whole population together. It would be more meaningful to consider other metrics that measure the Gaussianity of the distribution.

17. p 11, L326-333: Given the points 10, 11, and 16, I'm not fully convinced by the claim that the algorithm produces "accurate error estimates" and that it is as skilful as the authors claim in identifying areas where the derived Motion Vectors are less skilful. There is some skill improvement compared to assigning a single value, but that is a very low baseline to compare the results with. Quality Indicator values are, for instance, used at some NWP centres to assign situation-dependent observation error values to AMVs. How would the present algorithm compare to such a scheme? Also, the algorithm appears to perform not particularly convincingly in a situation where the truth was available for training and no measurement noise or retrieval errors further complicate the situation. How much skill will remain if it has to deal with these issues?

18. Fig. 7 and Fig. 10: The scale of the y-axis is rather large. The region of interest is probably confined to values < 20 m/s.

---

## Author Response (AR1)

**"Using Machine Learning to Model Uncertainty for Water-Vapor Atmospheric Motion Vectors"**

Teixeira et al.

Responses to Referee 1

We would like to thank the referee for the careful read of the paper and for the detailed comments. Please see our responses below:

 I have a problem with the present title. Reading it the first time I thought that paper was about improving error/quality of AMVs during the extraction process, and not during the assimilation process. From my understanding a title like: 'Use of Machines Learning to improve Uncertainty Quantification of Atmospheric Motion Vectors assimilated in NWP models', would certainly match better the real content of the paper and be less confusing.

We certainly understand the reviewer's outlook on this; uncertainty quantification can often be a confounding term with different interpretations across subject areas. The title has been modified to "Using Machine Learning to Model Uncertainty for Water-Vapor Atmospheric Motion Vectors" to reflect this.

2. The test presented in this paper is limited to water vapour AMVs extracted on specific layers. This potentially corresponds to extraction of 3D winds from hyperspectral sounders, as mentioned in the introduction. However, there is actually no evidence that the results can be generalised to the common AMVs extracted from clouds tracking in infrared or visible channels. If the method is limited to hyperspectral winds, this must be clearly specified in the text and probably also in the title of the paper, and not let the reader supposed that it works for all types of AMVs. If the method is not limited to hyperspectral AMVs authors have to present results also with common cloud motion winds extracted from satellite imagery. I understand from the text that another paper is upcoming (line 325), but there is no description or information that can actually let me assume that common AMVs have been used, and that the results are positive.

Referee #2 expressed similar concerns, and we can understand the reviewers' perspective. We have qualified our statement that the approach may be globally applicable to any measurements, and have stated more specifically that it is likely to be useful for other sources of AMVs (especially those obtained by tracking gradients in trace gases). In paragraph 3 of the introduction we have included additional mention of the height assignment errors known to be an issue in tracking cloud features from radiances. This source of error is expected not to be as great of an issue when tracking retrieved trace gases (as shown in Posselt et al. 2019), as it is when tracking cloud features or radiance images. In addition, there are sources of error that are expected to be common to any feature tracking algorithm (e.g., regions

without strong gradients in the field being tracked, or regions in which the wind is oriented parallel to contours in the field being tracked). We have modified our conclusions to include this in the last paragraph of discussion.

3. The algorithm seems to be too dependent on the user's choice of the number of clusters, and the paper does not discuss the dependence of the algorithm on the chosen training dataset. It is also very unclear if the different clusters identified could refer to kind of physical or geographical AMVs properties, or if they are only blindly resulting out of the numerical tests. Authors must clarify/discuss if the results may depend on the AMV extraction model used (Mueller 2017). It is not clear if the same clusters can be used for operational AMV extracted from other schemes too (NOAA, EUMETSAT, C2 JMA. . . Etc). If it is not the case I guess this study must be repeated individually for every different AMV extraction schemes and maybe after every releases of these codes, which should represent an important limitation for operational use in NWP models. Although the authors promise the possibility to distinguish different geophysical regimes, the application ultimately presented by the paper comes down to discriminating the AMVs that are null because they are tracking the ground radiance, which is much too simple to showcase the real benefits of the algorithm.

This study is meant to be a proof of concept – to show how a combination of random forest, plus a Gaussian mixture model, can be used to learn error structures found via comparison of simulated measurements with a reference "truth" dataset (as was done in our previous work). Naturally, the particular algorithm developed in this paper is wholly dependent both on the nature run and the AMV extraction method. However, it is not intended to be an algorithm that can be immediately used in NWP models. Instead, we aim to present a model that can be reproduced (and tuned) for use in specific contexts of AMV methods and data assimilation frameworks. The computational costs of training the algorithm (~1 day on a single processor, per pressure level) and even the computational costs of running the AMV extraction on the nature run (an average of 3 days per pressure level, on a non-optimized cluster network), are not outside the usual demands when updating parameters of NWP models.

In regards to the physical and geographical properties of the identified clusters, we have added a section in lines 329-345 and Figures 8-11 discussing this. They illustrate that the clustering algorithm manages to generally discriminate among geophysical regimes. Regarding the choice of number of clusters, this is a tuning parameter that is highly specific to application. We note that having one or more tuning parameters is not uncommon in many data analysis methods (e.g., k-means, PCA, self-organizing network, random forest, neural nets, regularized regression, smoothing splines, wavelets, etc.). Here, our method requires only 1 major tuning parameter (the random forest model also has tuning parameters, but that process, being a supervised regression, can be guided by cross validation). We note that the search for the 'optimal' number of clusters should be guided by expert knowledge, although this process should be greatly simplified by including an information criterion (e.g., the Bayesian Information Criterion) in the Gaussian Mixture Modelling algorithm. We have updated the end of the last paragraph of Section 3.4 to include this discussion.

Specific Comments:

1) Everywhere I would change the denomination "true wind" to "G5NR wind" throughout the text. No matter the quality of any dataset relating to physical quantities, it does not deserve to be called "true".

We understand that the term 'true' can often be controversial even when referencing a simulation. The denomination has been changed to 'Nature Run Wind' throughout the text. Thank you for the comment.

2) Line 144 It would be good to recall that this Figure relates to the first 1.5 months of the dataset, in the caption of the Figure.

Thank you. The distinction between training and test dataset has been made throughout the figure captions.

3) Lines 144-145 This is disappointing. Given the use of a powerful tool like GMM and the possibility of identifying "geophysical regimes" (line 132), I expected far more than just discriminating two groups, one being functional AMVs, and the other merely being the AMVs tracking the ground radiance, when the water vapour layer is too thin.

Figure 8-11, and lines 329-345 show that the clustering algorithm performs adequately in capturing consistent geophysical regimes. We focus in this paper on the 'skillfull' vs 'unskillfull' distinction because it is the most straightforward analysis for our purposes. More specific regime dependent uncertainties (as discussed in response to reviewer 2) is certainly a forward step after scaling this methodology beyond proof of concept.

4) Line 270 This parts misses a "is" between "xi" and "the".

Thank you for catching this. It was been corrected.

5) Section 4 The term Continuous Ranked Probability Score should be mentioned at least once before the formula at line 278. The two acronyms CPRS and CRPS are used in this section. Please correct.

The typo has been corrected. We mentioned the full name for CRPS immediately preceding its equation in (4), and we added a reference to a paper (Gneiting and Katzfuss, 2014) immediately after the equation.

6) Line 309 You are referring to Figure 13, and not Figure 12 as written.

Thank you for catching this. It has been corrected.

7) Lines 329-330 I find your conclusion a little daring, knowing that you had to try different numbers of clusters before actually managing to discriminate the null AMVs.

We apologize for the ambiguity. Our intention in these lines was different from what came across. We meant to say that our algorithm is able to 'find' or separate geophysically meaningful clusters without requiring domain knowledge expertise or prior information on the distribution of the variables. Granted, the algorithm requires the users to slide the number of clusters across some scales, but this process is vastly simplified since there is only 1 scalar parameter to vary. As we noted before, having tuning parameters is par-the-course for the majority of data analysis methods such as k-means, PCA, self-organizing network, random forest, neural nets, regularized regression, smoothing splines, wavelets, etc.

We understand the referee's concern, however. Therefore we have removed the aforementioned lines in the Conclusion, and we have included a note about the need to optimize over the number of clusters in  $2_{nd}$  paragraph of the Conclusion.

**"Using Machine Learning to Model Uncertainty for Water-Vapor Atmospheric Motion Vectors" Teixeira et al.**

Responses to Referee 2

We would like to thank the referee for the careful read of the paper and for the detailed comments. Please see our responses below:

1. My main criticism of the study is that I am unsure about the practical applicability of the results. The study relies on the "truth" being available from a nature run to train the algorithm in the first place (e.g., to derive the clustering, to derive the random forest). It is unclear to me how this will be circumvented for real-life applications, without introducing other problems that may jeopardise the performance of the algorithm. I am not convinced that the algorithm could be applied "as is" on Motion Vectors derived from humidity fields retrieved from real sounding data, and indeed no attempt is presented in the paper to investigate this. The paper should discuss how it is envisaged that the algorithm can be applied to real-life situations and what the potential problem areas are.

This study is meant to be a proof of concept – to show how a combination of random forest, plus a Gaussian mixture model, can be used to learn error structures found via comparison of simulated measurements with a reference "truth" dataset (as was done in our previous work). As such, we would not expect the results to be applicable "as is". However, we do expect that there are certain errors endemic to AMVs that are captured by our algorithm, and as such are also applicable to other scenarios. We have revised our conclusions to contain a discussion of this issue, but in summary we expect that current practice in numerical weather prediction may provide guidance here. While we never know "truth" in any practical application, there are ways to approximate errors without having exact knowledge of the true field. This is done routinely to characterize errors in any observation used in any data assimilation system. Typically, error estimation involves comparison with respect to an independent dataset, and in the case of our machine learning algorithm, a similar procedure could be followed.

Furthermore, we note that in this paper we are primarily interested in the distribution of a retrieved quantity versus the hidden truth. That is, given a retrieved value  $\hat{Y}_i$ , we are interested in the first and second moments (i.e., E( $\hat{Y}_i - Y$ ) and var( $\hat{Y}_i - Y$ ))). We model our uncertainty *relative to the truth*, and therefore we cannot avoid the need to have some instances of the true data, or proxies thereof. This is a departure from much of the literature on uncertainty modelling with machine learning (e.g., Coulston et al., 2016; Tripathy et al., 2018; Tran et al., 2019; Kwon et al., 2020), which primarily define the uncertainty of a prediction as var( $\hat{Y}_i$ ), or how sensitive that prediction is to tiny changes in the models/inputs. Our methodology allows for error estimates that fit naturally within the data assimilation framework, and, unlike the sensitivity estimate var( $\hat{Y}_i$ ), also enable hypothesis testing and risk determination in support of decision making. To address the referee's concern, we have expanded on this in the 2nd paragraph of Section 3.1 and the 4th paragraph of the conclusion.

2. In several areas the manuscripts appears to suggest that the method would be generally applicable, ie to other AMVs and possibly beyond (e.g., p3 L80 "... our methodology in principle could be used to quantify uncertainty in any measurements..."). I think this should be qualified. Subject to the point above, the algorithm may offer some value for AMVs derived from sounder retrievals; I suspect the value for the cloud-tracked AMVs is very limited - though these are currently the most widely used AMV datasets. There may be applicability beyond this, but the authors should explain more clearly how they expect the algorithm to be applied to "any measurement".

We have qualified our statement that the approach may be globally applicable to any measurements, and have stated more specifically that it is likely to be useful for other sources of AMVs. There are sources of error that

are expected to be common to any feature tracking algorithm (e.g., regions without strong gradients in the field being tracked, or regions in which the wind is oriented parallel to contours in the field being tracked). We have modified our conclusions to include this discussion.

3. It would be useful if the authors took a critical look at the physical basis or motivation of their algorithm. The algorithm attempts to provide an uncertainty estimate for a derived wind vector with the derived wind vector and water vapour as the only inputs. I would expect other factors to play a considerable role, such as predictors describing the texture of the scene (to characterise the likely success of the tracking step), or C2 predictors that describe more the meteorological conditions (to characterise how likely humidity features are passive tracers). Spatial consistency measures such as the ones typically used in the formulation of the Quality Indicator (Holmlund 1998) may also be relevant. The predictor choice used in the study appears ad-hoc to me, and it could almost certainly be improved.

The predictor choice is indeed constrained and could almost certainly be improved in implementation. However, the limits on the input variables are a specific decision and not an oversight. The framework presented in this paper is not to necessarily intended to produce the best possible AMV uncertainty algorithm but to show, in a proof of concept, what a purely data-driven approach can lead to. In particular, we based our approach around the state-dependent errors characterized in Posselt et al. (2019), and sought to build an error characterization model that is itself state-driven. Including other parameters towards improving the algorithm is certainly interesting, and would most likely occur when implementing this methodology at scale, but is beyond the specific intentions of this paper. We address this in L250-257. Furthermore, we see in Figures 8-11 that even these limited inputs can produce physically recognizable regimes.

**Specific points:**

1. Title: I find the title misleading, as the authors only address the uncertainty in the wind estimates, not the height assignment uncertainty, which is a leading contributor of uncertainty for the most commonly used AMVs. The use of "Atmospheric Motion Vectors" may also lead readers to believe they will read about cloudy-tracked winds, when the links to these in the manuscript are very weak. I suggest to be more specific in the title, maybe "Estimation of uncertainty in wind retrievals derived from tracking humidity structures using Machine Learning".

We understand the reviewer's viewpoint on this. With additional consideration of the comments from reviewer 1 on the subject matter, the title has been rewritten to reflect that the paper concerns water-vapor AMVs. With vapor-vapor AMVs, height assignment uncertainty is less of a concern (we address this in our response to specific point #3), and should guide the reader towards a better interpretation of what the paper covers.

2. p2, L34: Nguyen et al (2019) is referred to quite extensively in the paper (here and elsewhere), but is listed as a comparatively inaccessible report from the National Institute for Applied Statistics Research Australia. A journal paper with a similar title has recently been published, and I wonder whether this could be referred to instead.

The reference noted has been replaced with the most updated reference to this paper.

3. p2, L 44-45 "However, height assignment is not the dominant portion of the error. . . .": This is a strong claim to make, and I think it needs to be backed up with a suitable reference. Retrievals from infrared or microwave sounders do not represent radiosonde-like profiles. For a given level in the retrieved profile, the averaging kernel will describe the characteristics in the vertical represented by the retrieval - and these are not Diracdelta functions. Height characteristics of AMVs derived from such retrievals will hence be rather complex, and interpreting them subsequently as single-level winds may well be a considerable contribution to the error budget. I am not aware that this aspect has been thoroughly investigated in the literature yet. It should at least be mentioned in the present study.

This statement has been rephrased and expanded upon in lines 54-57. We acknowledge that height assignment error due to misspecification of height in the water vapor profiles could be impactful on the uncertainties for the extracted AMVs. However, this uncertainty cannot be directly assessed through analysis of the AMV extraction algorithm alone. Instead, it necessitates quantified uncertainties on the water vapor profiles themselves, something which is well beyond the scope of this paper.

4. p2, L51 "The Expected Error . . . to correct AMV observation error.": The EE aims to provide an estimate of the statistical characteristics of the observation error, but does not try to correct any errors in the AMVs. Please rephrase.

Thank you for noticing this. This has been rephrased as recommended.

5. p3, L90/91: It would be useful to provide an idea of the spatial scales used in the tracking step, ie what is the typical size of the target used.

The tracking step size is a 33km grid box for a sigma level of 4.2. More details can be found in Posselt et al (2019).

6. p 3, L 100/101, Fig. 1: The authors emphasise the poorer performance in drier regions. While it is a little harder to see, my impression is that there is also poorer performance near frontal features (e.g, positive biases East of South America or East of North America). Poorer performance around frontal regions seems physically plausible, as single-level humidity may not be a passive tracer in these regions. I think it would be worth commenting on this in the main text. This could also motivate a predictor other than water vapour in the scheme developed later.

We also suspect that vertical motion may be part of the reason behind the large errors near fronts, although a portion is also certainly due to the features identified in Posselt et al. (2019) (winds oriented along lines of constant water vapor). This paper aims to model uncertainties that are both regime dependent and state dependent. Obviously, these are intertwined: we see in figure 11 that the unskillful cluster 6 has a large representation on the east coasts of North and South America, indicating that it is at least partially capturing this frontal dynamic. When optimizing the methodology at scale, special consideration for more specific regime types (that are not purely state dependent) is a positive way of improving the uncertainty modelling approach for specific needs.

7. p 5, L139-142: It is not quite clear to me whether the description of the training/testing dataset in this paragraph is effectively referring to the same datasets described later (p8 L248/249). I got the impression here that all data for the 1.5/0.5 months were used, but later it sounds as if the dataset was subsampled. I suggest making this clearer to avoid confusion.

The text has been rewritten to make it clearer that data has been subsampled from the training and testing datasets.

8. p 6, L187-191: It would be good if the authors could motivate further how they chose 9 clusters in the Gaussian mixture model. The text sounds as if it was a subjective choice, but maybe there was an objective component as well? Given the very limited inputs to characterise the conditions, and the lack of clear distinctions between some clusters, the chosen number of clusters appears high.

As the reviewer suspected, there was a combination of quantitative and qualitative reasoning in determining the number of clusters. We address this in lines 329-345. New figures 8-11 also show greater clarity of the distinction between clusters.

9. p 7, L224/225 "Relative to . . . entire dataset.": I am unsure about what is meant here. I suggest rephrasing.

This redundant sentence has been removed.

10. p 8, first paragraph: It looks to me as if the clustering algorithm performs significantly more poorly once the true wind value has been substituted. Contrary to what is said in the text, clusters 4 and 5 shown in Fig. 9 appear relatively unskilful, certainly in comparison to the same clusters shown in Fig. 6. Also, it looks as if the population in clusters 6 and 8 (referred to as the "unskilled" regimes) is very low, and much lower than what was found in Fig. 6. It appears that the assignment into these clusters is very different to what was possible before. This may not be too surprising, as the previous assignment had the benefit of the truth being available, but the aspect is not addressed much in the text.

This is certainly a chief concern we have with the approach. There is substantial degradation in the clustering algorithm's performance when the model is not given the true winds. An implementation of this methodology at scale could benefit from an improvement in the random forest (or its replacement with a better performing emulator). This is addressed in lines 261-266. We must note, however, that our ultimate intention is not to create a machine learning emulator for the wind-tracking algorithm, but simply to employ it to model uncertainties.

11. p8, second paragraph/Fig. 11: Are the differences in standard deviation or bias between the clusters statistically significant? Also, what is the relative population of each cluster? Judging by Fig. 9 and 10, the clusters with the most different standard deviation (clusters 6 and 8) appear to have relatively small populations, whereas the variation in standard deviation in the remaining clusters is smaller.

We have over 800,000 observations in the dataset, and their relative population is listed below

| Regir | ne Cou | int Percent |
|-------|--------|-------------|
| 1     | 42308  | 4.95%       |
| 2     | 77545  | 9.08%       |
| 3     | 49187  | 5.76%       |
| 4     | 231268 | 27.07%      |
| 5     | 190543 | 22.31%      |
| 6     | 311    | 0.04%       |
| 7     | 206353 | 24.16%      |
| 8     | 41223  | 4.83%       |
| 9     | 15491  | 1.81%       |
|       |        |             |

To address the question of whether the differences in standard deviation (std) or bias between cluster is statistically significant, we opted to construct confidence intervals for the bias and std within each regime using the bootstrap (Efron and Tibshirani, 1993). The procedure of our bootstrap is as follows

- 2. Sample *with replacement* Nj observations from this subset. This forms a bootstrap sample
- 3. From 2., compute an estimate of the bias and std.
- 4. Repeat step 2-3 for 1000 times, giving us 1000 estimates of the bias and 1000 estimates of the std within regime j.
- 5. Compute 95% confidence intervals from the 1000 estimates of bias and std from 4.

The results for the confidence intervals (in graphical forms) are listed below:

Figure: Top rows (bias and std confidence intervals for u-wind), bottom rows (bias and std confidence intervals for v-winds). The interval represent a 95% confidence interval.

We note that the Figure above indicates that for many of the biases, they can be considered unbiased since their confidence interval includes 0 (e.g., regimes 2-8 for u-wind). However, the plot also clearly indicates that two regimes are statistically different from 0 (regime 1 and 9). We also note that for the standard deviation maps, the CI's indicate that they are fairly stable (small narrow range) and that most of the regimes have statistically different standard deviation (denoted here visually as CI's that do not overlap one another). We also note that u and v wind direction tend to have very similar patterns, indicating that our regime classification is persistent across u and v.

To summarize, the CI plot above indicate that the differences in std between different regimes are highly statistically significant (as evidenced by the small confidence intervals and their spacing). For the biases, 3 of the regimes are statistically significantly different from the rest (i.e., regimes 1, 6, and 9), while the rest are likely relatively unbiased (i.e., bias = 0).

12. p 8, L248/249: The authors mention that they use a training set of 1,000,000 points, and a testing dataset of the same number of points. How have these been chosen within the available data? It looks as if many more points were available, at least for the training dataset. Also, the link to p 5 L139-142 was not quite clear to me.

We apologize for the lack of clarity on line 239-242. What we meant was that we used the NatureRun data from Posselt et al. (2019), which applied an AMV algorithm to outputs from the NASA Goddard Space Flight Center (GSFC) Global Modeling and Assimilation Office (GMAO) GEOS-5 Nature Run (G5NR; Putman et al. 2014). The

Nature Run is a global dataset with ~7 km horizontal grid spacing that includes, among other quantities, threedimensional fields of wind, water vapor concentration, clouds, and temperature. The AMV algorithm is applied on four pressure levels (300hPa, 500hPa, 700hPa, and 850hPa) at 6-hourly intervals, using three consecutive global water vapor fields spaced one hour apart, and for a 60-day period from 07/01/2006 to 08/30/2006. In this paper, we make use of this dataset, although we focus only on the data at 700 hPa. We updated the manuscript on line 123-124 to refer to the data description in Section 2.1 and to make clear that we are using the data at 700 hPa.

Regarding the full dataset, it uses a 5758 x 2879 grid for longitude and latitude, with 240 time steps (60 days at 6 hours intervals). This forms a 5758 x 2879 x 240 = 3978547680 data points, which is simply too large for us to feasibly train a model. Therefore, we subsampled 1,000,0000 data points from this dataset uniformly where each of the 3978547680 data point has an equal chance of being selected.

Thank you for bringing this point to our attention. We have clarified the paper about the sampling process on the  $1_{st}$  bullet point of Section 3.7, and we have added information about the lon, lat, time grid at the bottom of the  $1_{st}$  paragraph in Section 2.1.

**13. p 9, formula 4 and elsewhere: Typo: CPRS should be CRPS.**

Thank you for catching this. The typo has been fixed.

14. p 9, L279-283: The " $\leq$ " in L282 appears to be inconsistent with what is said about CRPS earlier in the paragraph.

Thank you for catching this. The mistake was earlier in the paragraph, and has been addressed.

15. Fig. 12 and 13: Are these showing results for the test dataset? I assume they do (based on what is said on p 5, L141/142), but I think it would be clearest if this information was provided in the caption (a similar comment could be made for Fig. 6-11).

The distinction between the training and test dataset has been made throughout the figure captions.

16. p 10, L306-311: The authors point to the finding that the residuals normalised with the estimated error have a standard deviation close to 1. It's a useful cross-check, but I suspect this finding primarily reflects that the training and testing data has similar standard deviations of AMVs vs true winds. I suspect it would have been obtained by assigning one constant observation error equal to the standard deviation of the whole population together. It would be more meaningful to consider other metrics that measure the Gaussianity of the distribution.

The reviewer's assessment is correct in that assigning a constant observation error equal to the standard deviation of the whole population together would also produce normalized residuals with standard deviation close to 1. However, this test is designed to show that our error predictions are actually consistent with the variability in validation data (this is termed 'validity' in the statistical literature).

As a thought experiment, consider the case of optimal estimation uncertainty estimates. Optimal estimation (Rodgers, 2000) purports to make estimates of the distribution  $[\hat{Y}_i - Y]$  by making assumptions about the data structures, distributions, and/or forward models, and the robustness of these uncertainty estimates are usually only valid if these assumptions are correct. It is well-known in remote sensing that retrievals from optimal estimations tend to produce uncertainty estimates that are too low relative to validation data (Hobbs et al., 2017). For example, the uncertainties from OE for the Orbiting Carbon Observatory-2 (OCO-2) instrument tend to be too small (relative to validation data) by a factor of two. If we applied the same z-score test to the OCO-2 data, we would have obtained standard deviations of z-scores that is probably around 2, indicating that there is something awry with their error estimates.

The referee has noted that an error estimate can be 'valid' without being useful (this is the case with using the population standard deviation). This is why we also included the discussion on the CRPS, which gives a

comparative assessment of skill (or usefulness) between two different predictions, and we have shown in this paper that our regime-based method is more skillful than using the population-based mean, and at the same time its error predictions are also valid.

17. p 11, L326-333: Given the points 10, 11, and 16, I'm not fully convinced by the claim that the algorithm produces "accurate error estimates" and that it is as skilful as the authors claim in identifying areas where the derived Motion Vectors are less skilful. There is some skill improvement compared to assigning a single value, but that is a very low baseline to compare the results with. Quality Indicator values are, for instance, used at some NWP centres to assign situation-dependent observation error values to AMVs. How would the present algorithm compare to such a scheme? Also, the algorithm appears to perform not particularly convincingly in a situation where the truth was available for training and no measurement noise or retrieval errors further complicate the situation. How much skill will remain if it has to deal with these issues?

We understand the reviewer's concerns in this regard. The uncertainties presented in this paper are not, in it of themselves, a marked improvement from state of the practice AMV uncertainty modelling. But neither are they intended as such; this paper is a proof of concept for the methodology it entails. As discussed in previous comments, there is no doubt that the algorithm itself can be tuned and enhanced for specific use cases. This would involve some reckoning with retrieval error of the water vapor features. However, we do believe that the paper demonstrates that even a bare-bones implementation of the approach can produce uncertainties that are valid and, to some degree, useful. We further note that they are produced in a physics-agnostic framework with no underlying assumptions and, critically, with a data-driven analysis of only the state elements. The research presented in Posselt et al. (2019) is fundamental in driving the analysis in this paper: state-dependent errors provide the context for a purely state-dependent uncertainty modelling approach. Ultimately, we hope to add to the literature and understanding of AMV uncertainty modelling, not supplant existing approaches. To the extent that the specific uncertainties produced in this paper are useful, that will be exhibited in an upcoming paper.

Fig. 7 and Fig. 10: The scale of the y-axis is rather large. The region of interest is probably confined to values

4 1Jet Propulsion Laboratory, California Institute of Technology

5 Abstract. Wind-tracking algorithms produce Atmospheric Motion Vectors (AMVs) by tracking clouds or water vapor 6 across spatial-temporal fields. Thorough error characterization of wind-tracking algorithms is critical in properly 7 assimilating AMVs into weather forecast models and climate reanalysis datasets. Uncertainty modelling should yield 8 estimates of two key quantities of interest: bias, the systematic difference between a measurement and the true value, 9 and standard error, a measure of variability of the measurement. The current process of specification of the errors in 10 inverse modelling is often cursory and commonly consists of a mixture of model fidelity, expert knowledge, and need 11 for expediency. The method presented in this paper supplements existing approaches to error specification by 12 providing an error-characterization module that is purely data-driven and requires few tuning parameters. Our 13 proposed error-characterization method combines the flexibility of machine learning (random forest) with the robust 14 error estimates of unsupervised parametric clustering (using a Gaussian Mixture Model). Traditional techniques for 15 uncertainty modeling through machine learning have focused on characterizing bias, but often struggle when 16 estimating standard error. In contrast, model-based approaches such as k-means or Gaussian mixture modelling can 17 provide reasonable estimates of both bias and standard error, but they are often limited in complexity due to reliance 18 on linear or Gaussian assumptions. In this paper, a methodology is developed and applied to characterize error in 19 tracked-wind using a high-resolution global model simulation, and it is shown to adequately capture the error features 20 of the tracked wind.

**21 1. Introduction**

22 Reliable estimates of global winds are critical to science and application areas, including global chemical transport 23 modeling and numerical weather prediction. One source of wind measurements consists of feature-tracking based 24 Atmospheric Motion Vectors (AMVs), produced by tracking time sequences of satellite-based measurements of 25 clouds or spatially distributed water vapor fields (Mueller et al., 2017; Posselt et al., 2019). The importance of global 26 measurements of 3-dimensional winds was highlighted as an urgent need in the NASA Weather Research Community 27 Workshop Report (Zeng et al., 2016) and was identified as a priority in the 2007 National Academy of Sciences Earth 28 Science and Applications from Space (ESAS 2007) Decadal Survey and again in ESAS 2017. For instance, wind is 29 used in the study of global CO2 transport (Kawa et al., 2004), numerical weather prediction (NWP; Cassola and 30 Burlando, 2012), as inputs into weather and climate reanalysis studies (Swail and Cox, 2000), and for estimating 31 current and future wind-power outputs (Staffell and Pfenninger, 2016).

32 Thorough error characterization of wind-track algorithms is critical in properly assimilating AMVs into forecast 33 models. Prior literature has explored the impact of 'poor' error-characterization in Bayesian-based approaches to Deleted: Quantification

| (  | Deleted: (also known as uncertainty quantification) |
|----|-----------------------------------------------------|
| (  | Deleted: quantification                             |
|    |                                                     |
| (  | Deleted: input into                                 |
|    |                                                     |
| (  | Deleted: methods                                    |
| -( | Deleted: supplement                                 |
| ~( | Deleted: This paper proposes an                     |

[revised manuscript text omitted]
 150 corresponding contributing factors) section 3.{ INCLUDEPICTURE (and their in

151 "https://lh4.googleusercontent.com/Bxx2AuV0Yv\_LyfydtO30hkD9PeGug6p\_AMNp7hKH4ZIU9SY-

152 rZBzlPepaT-fAG51TilWVrFM0KlfBkBZLjfWQbubq8aSFsxKvRu0LGALEH-

153 cNQpJeJ1qvxE6Dimat5t6hP2UFfCK" \\* MERGEFORMATINET }

154 2.2 Importance of Uncertainty Representation in Data Assimilation

155 Proper error characterization for any measurement, including AMVs, is important in data assimilation. Data

156 assimilation often uses a regularized matrix inverse method based on Bayes' theorem, which, when all probability

157 distributions in Bayes' relationship are assumed to be Gaussian, reduces to minimizing a least-squares (quadratic) cost

158 function Eq (1):

$$J = (x - x_b)B^{-1}(x - x_b) + ((y - a) - H[x])^{1}R^{-1}((y - a) - H[x])$$

160 where x represents the analysis value, xb represents the background field (first guess), B represents the background 161 error covariance, y represents the observation, and H represents the forward operator that translates model space into Deleted: The

| ( | Deleted: true |
|---|---------------|
|   | Deleted: true |
| ( | Deleted: true |
| ( | Deleted: true |
| ( | Deleted: are  |

| λ     | Formatted: Font: Bold |
|-------|-----------------------|
| /     | Formatted: Font: Bold |
| (     | Formatted: Font: Bold |
| (     | Formatted: Font: Bold |
| ····( | Formatted: Font: Bold |

{ PAGE }

(1)

[revised manuscript text omitted]

| Deleted: is capable of determining whether any                                                  |
|--------------------------------------------------------------------------------------------------------|
| Deleted: belongs in the 'skilled' or 'un-skilled' cluster                                              |
| Deleted: , since we see the relationship between the error-
regimes and water vapor content. |
| Deleted: a                                                                                             |

255 3. Model the distribution of the population  $P(\theta)$  as:

256
$$P(\theta) = \sum_{j=1}^{K} \pi_{j} N(\mu_{j}, \Sigma_{j})$$
(3)

257 Where  $N(\mu_j, \Sigma_j)$  is the normal distribution with mean  $\mu_j$  and covariance  $\Sigma_j$  of the *j*-th cluster.

258 K is the number of clusters, and  $\pi_j$  is the mixture proportion.

259 4. Determine  $\pi_i, \mu_i, \Sigma_i$  for K clusters using the Expectation–Maximization Algorithm

260 5. From 3. and 4., estimate the probability of a given  $x_i$  belonging to the j-th cluster as  $P(x_i \in k_j) = p_{ij}$

261 6. Assign point xi to the cluster with the maximum probability pij

 262
 The mixture model clustering is based on the R package 'Mclust' developed by Fraley et al. (2012), which builds upon

 263
 the theoretical work of Fraley and Raftery (2002) for model-based clustering and density estimation. The process uses

 264
 an Expectation-Maximization algorithm to cluster the dataset, estimating a variable number of distinct multivariate

 265
 Gaussian distributions from a sample dataset. Training the Gaussian mixture model on this dataset provides a

 266
 clustering function which outputs a unique cluster for any data point with the same number of variables.{

 267
 INCLUDEPICTURE

 268
 "https://lh5.googleusercontent.com/hTyZLCheUt01nax60CT

 268
 PrH\_M3\_pNyiVsopj3jyTJr8CNcKeTT\_GkQa80dAOLFIWvxtdQ9EApaE5c8G2WyjIYfk1ihxPSxClrO5xtAz0LgG2

269 ToHP00myCbV6YGIEMXnwpn1FE5n6" \\* MERGEFORMATINET }

. . .

270 In one dimension, a Gaussian mixture model looks like the distributions depicted in { REF \_Ref29398417 \h \\*

271 MERGEFORMAT }: instead of modeling a population as a single distribution (Gaussian or otherwise), the GMM 272 algorithm fits multiple Gaussian distributions to a population. One key aspect of this algorithm is the capability of 273 assigning a new point to the most likely distribution. For example, in the 1-D figure, a normalized AMV estimate with 274 a value of 10 would be more likely to originate from the broad cluster '2' than the narrow cluster '4'. In this case, we 275 model the population as a Gaussian mixture model in five-dimensional space, which consists of two Nature Run wind 276 vector components (u and v), two AMV estimates of these wind components (u and v), and the simulated water vapor 277 values, all of which have been standardized, to have mean 0 and standard deviation of 1. Each cluster has a 5-278 dimensional mean vector for the center and a 5x5 covariance matrix defining their multivariate Gaussian shape. The 279 estimation of a covariance matrix allows for the characterization of the relationships between the different dimensions 280 within each cluster, and as such the gaussian mixture model approach provides greater potential for understanding the 281 geophysical basis of error regimes than other unsupervised clustering approaches,

- be achieved by including additional meteorological or geographic information. However, the intention of this paper is to study the ability of a purely data-driven approach, where no additional information or assumptions are passed to
- the machine learning model outside of the inputs and outputs to the AMV algorithm itself. Posselt et al. (2019) showed

[revised manuscript text omitted]

| 505
506
507
508
509
510 | The relative behavior of the CRPS is consistent between u and v winds. The CRPS tends to have to wider distribution when applied to the regime-based error characterization. Compared to the alternative error characterization scheme, our methodology produces a cluster of highly accurate predictions (low CRPS scores), in addition to some cluster of very uninformative predictions (high CRPS scores). These clusters correspond to the highly skilled cluster (e.g., Cluster 3) and the unskilled clusters (Cluster 6 and 8), respectively. Overall, the mean of the CRPS is lower for our methodology than it is for the alternative method, indicating that as a whole our method produces a more accurate |                           | Deleted: likely
|----------------------------------------|-----------------------------------------------------------------------------------------------------------------------------------------------------------------------------------------------------------------------------------------------------------------------------------------------------------------------------------------------------------------------------------------------------------------------------------------------------------------------------------------------------------------------------------------------------------------------------------------------------------------------------------------------------------------------------------------------------------------------|---------------------------|----------------------------------------------------------------------------------------|
| 511                                    | probabilistic forecast.                                                                                                                                                                                                                                                                                                                                                                                                                                                                                                                                                                                                                                                                                               |                           |                                                                                        |
| 512
513
514
515
516        | Thus far we have shown that our method produces more accurate error-characterization than an alternative method based on marginal means and variance. Now, we assess whether our methodology provides valid probabilistic prediction; that is, we test whether the uncertainty estimates provided are consistent with the empirical distribution of the validation data. To assess this, we construct a metric in which we normalize the difference between the Nature Run wind and the tracked wind by the predicted variance. That is, for the i-th observation , we compute the normalized                                                                                             |                           | Deleted: true                                                                          |
| 517                                    | values for $\mu_i$ and $\gamma_i$ using the following equations:                                                                                                                                                                                                                                                                                                                                                                                                                                                                                                                                                                                                                                                      |                           | Deleted: u                                                                             |
|                                        |                                                                                                                                                                                                                                                                                                                                                                                                                                                                                                                                                                                                                                                                                                                       | - Andrews                 | Deleted: v                                                                             |
| 518                                    | $z_{u,i} = \frac{u_i - u_i}{\sigma_{u,i}}$                                                                                                                                                                                                                                                                                                                                                                                                                                                                                                                                                                                                                                                                            |                           | $\begin{array}{l} \textbf{Deleted: } z_u = \frac{u-u}{\sigma_u} \\ \\ z_v \end{array}$ |
| 519                                    | $z_{v,i} = \frac{v_i - v_i}{\sigma_{v,i}} $ (5)                                                                                                                                                                                                                                                                                                                                                                                                                                                                                                                                                                                                                                                            |                           |                                                                                        |
| 520                                    | Where $\mu_i$ is the i-th Nature Run u wind from the Nature Run data, $\mu_i$ is the tracked-wind, and $\rho_{u,i}$ is the error as                                                                                                                                                                                                                                                                                                                                                                                                                                                                                                                                                                     |                           | Deleted: u                                                                             |
| 521                                    | assessed by our model (recall that it is a function of the regime index to which $\alpha_i$ has been assigned). The values for                                                                                                                                                                                                                                                                                                                                                                                                                                                                                                                                                                                        |                           | Deleted: true                                                                          |
| 522                                    | the v-wind are defined similarly. The residuals in Eq (5) can be considered as a variant of the z-score, and it is                                                                                                                                                                                                                                                                                                                                                                                                                                                                                                                                                                                                    | $\langle \rangle \rangle$ | Deleted: u                                                                             |
| 523                                    | straightforward to see that if our error estimates are valid (i.e., accurate), then the normalized residuals in Eq. (5)                                                                                                                                                                                                                                                                                                                                                                                                                                                                                                                                                                                               | $\langle \rangle$         | Deleted: $\sigma_u$                                                                    |
| 524                                    | should have a standard deviation of 1. If our uncertainty estimates $\sigma_{u,i}$ and $\sigma_{v,i}$ are too large, then the standard deviation                                                                                                                                                                                                                                                                                                                                                                                                                                                                                                                                                                      |                           | Deleted: .                                                                             |
| 525                                    | $\underline{of } z_{u,i} \underline{and } z_{v,i} \underline{should}$ be less than 1; similarly, if our uncertainty estimates are too small, then the standard deviation                                                                                                                                                                                                                                                                                                                                                                                                                                                                                                                                              |                           | Deleted: In { REF_Ref29398184 \h \* MERGEFORMAT }.                              |
| 526                                    | of z u,i and z v,i should be larger than 1. In { REF_Ref45710459 \h \* MERGEFORMAT }, we display the histogram                                                                                                                                                                                                                                                                                                                                                                                                                                                                                                                                                                                  |                           |                                                                                        |
| 527                                    | of the normalized residuals $z_u$ and $z_v$ . It is clear that for both types of wind, the standard deviation of $z_{u,i}$ and $z_{v,i}$ are                                                                                                                                                                                                                                                                                                                                                                                                                                                                                                                                                                          |                           |                                                                                        |
| 528                                    | 1.003 and 1.009, respectively, indicating that our error characterization model is highly accurate when forecasting                                                                                                                                                                                                                                                                                                                                                                                                                                                                                                                                                                                                   |                           | Deleted: methodology produces                                                          |
| 529                                    | uncertainties                                                                                                                                                                                                                                                                                                                                                                                                                                                                                                                                                                                                                                                                                                         |                           | Deleted: (std = 1.003 and 1.009 for u and v, respectively).                     |
| 530                                    | 5 Conclusion                                                                                                                                                                                                                                                                                                                                                                                                                                                                                                                                                                                                                                                                                                          |                           | Formatted: Font color: Black                                                           |
| 531                                    | Error characterization is an important component of data validation and scientific analysis. For wind-tracking                                                                                                                                                                                                                                                                                                                                                                                                                                                                                                                                                                                                        |                           | Deleted: Uncertainty quantification, which is the                               |
| 532                                    | algorithms, whose outputs (tracked u and v) are often used as observations in data assimilation analyses, it is necessary

---

## Author Response (AR2)

**"Using Machine Learning to Model Uncertainty for Water-Vapor Atmospheric**

**Motion Vectors"**

**Teixeira et al.**

**Responses to Referee 1**

We would like to thank the referee for the careful read of the paper and comments. Please see our responses below:

**GENERAL COMMENTS**

The second version of this paper is greatly improved. I would like to thank the authors for the substantial effort they made to answer referees concerns.

I especially appreciated the addition of the Fig8-11 that illustrate the physical and geographical properties of the identified clusters. The general presentation of the method as a 'proof of concept' also improved a lot the overall coherency of the paper with the actual results.

I can now accept this paper for publication.

We thank the reviewer for their detailed and helpful comments on the first review of the paper. We are glad to see that we have met the reviewer's acceptance for publication.

**SPECIFIC COMMENTS**

1) Line 228 'to the Nature Run winds within each cluster (),' Probably missing something inside the parenthesis.

Thannk you for noticing this. We have fixed it by including the figure reference needed.

**2) Line 274-275**

I do not understand the sentence: 'We trained a random forest with 50 trees on a separate set of tracked winds and water vapor values to predict Nature 274 Run winds using the 'randomForest' package in R.' What is 'R'?

'R' refers to the R programming language, a language used for statistical analysis. This has been noted in the paper.

**3) Line 380-381**

'In this paper we demonstrate the application of a machine learning uncertainty modeling tool to AMVs derived from hyperspectral sounder water vapor profiles.' This is not exactly correct. No data from hyperspectral sounders have been used in this study. Please precise or correct.

Thank you for this comment. This has been corrected to 'water vapor profiles intended to mimic hyper-spectral sounder retrievals.'

**"Using Machine Learning to Model Uncertainty for Water-Vapor Atmospheric Motion Vectors"**

**Teixeira et al.**

**Responses to Referee 2**

We would like to thank the referee for the careful read of the paper and for the detailed comments. Please see our responses below:

1. While the manuscript has improved in several areas, I feel that some of my earlier more substantive comments have not been sufficiently addressed. These include comments on the usefulness of the algorithm in practical situations (earlier general points 1 and partially 3), the robustness of the performance (particularly of the clustering; general point 1, specific points 6 and 10), and the evaluation of the performance (specific points 16 and 17). In some of their replies to the comments the authors acknowledge limitations and issues of their algorithm, yet this is not reflected in the revised manuscript (especially general point 3, specific points 6 and 10). I therefore continue to be unconvinced that this algorithm is useful for the intended application.

For the paper to be acceptable for publication, I think these points need to be more thoroughly addressed. In my view this means that the results are either more critically evaluated and the limitations more clearly outlined, or the algorithm has to be refined and made more robust so that applicability to real-life applications is indeed ensured. In my view either of this requires another major revision. I do not think that the present results serve as a "proof-of-concept" of a useful algorithm that could be applied to practical situations with real data, as the authors claim. I could accept if the paper was written from the perspective of introducing a novel conceptual framework which has been explored in an initial implementation, which shows some skill in assigning regime-specific observation errors, but has revealed a range of issues that would need to be addressed for this to be a viable and useful algorithm for real applications (and which may not be possible to address).

We thank the reviewer for their insightful comments. We agree with the reviewer that what we intend to present is more of a 'conceptual framework' than a 'proof-of-concept'. We never intended to present this work as a 'ready-to-go' algorithm in this particular implementation; instead, we laid out the foundation for an uncertainty modelling approach which we plan to implement at a larger scale in subsequent work. To summarize, the conceptual foundations to this framework are the following:

1. We intend to model uncertainty in the AMV algorithm relative to the underlying value it is trying to capture. As described in previous responses and detailed in the paper, this is a departure from most machine learning based approaches to uncertainty modelling.

- 2. These uncertainties, and further their relationship with state vector elements, have shown to discretize themselves into different error regimes. We focus our approach on characterizing these regimes.
- 3. Following the work by Posselt et al (2019), we believe that these uncertainties are state-dependent. As such, our framework explicitly intends to examine these uncertainties as function of the relationship between state values. This is not to say that implementations of this framework should exclude additional information; on the contrary, we believe that the addition of context-dependent information could greatly enhance an implementation of this framework. However, at its core, we attempt to model state-dependent uncertainties with a state-dependent model.
- 4. We believe that the most versatile framework, in terms of potential application, is one which at its base is context-agnostic. A purely data-driven approach, as we lay out in the paper, provides the platform for context-dependent tuning when scaling the methodology.
- 2. A clear path of how these issues could be addressed should be provided, including an outline how the algorithm could be derived without the truth being available . The latter requires recognition and discussion of the issues that will be encountered when substituting the truth with proxy data beyond the superficial suggestion that has been added in the conclusion section. These proxy data (the authors suggest reanalysis data or collocated reference observations) will introduce their own regime-specific errors, and there is no mechanism apparent in the present implementation of the algorithm that would be able to separate the errors in the proxy data (or the collocation errors), rather than in the AMVs. In addition, real-life applications would encounter errors in the humidity retrievals used for the tracking, and these will affect the characteristics of the errors in the tracked winds as well as the performance of the algorithm.

We introduce our framework in an environment that is limited and well-behaved, but which nonetheless we believe provides insight into how such an approach would perform at a larger scale. The uncertainty model shows skill in discerning regime specific uncertainties in this scenario. Of course, there are issues when moving from the controlled environment of the simulation study to large scale applications. We understand these to be: (1) the existence of uncertainty on the tracked humidity values, and (2) the ability of the training dataset to adequately capture both the range of conditions of water vapor and wind speed, and their inherent relationship. We have rewritten and expanded the conclusion in the paper to include a lengthier discussion of these topics, as well as recommendations on how to address them. As a reference, this includes:

- 1. A discussion of how humidity retrieval uncertainty would impact our model and approaches to address it. L417-431.
- A discussion of considerations future users must make when developing training data for the model. L432-466; L481-486.
- A list of specific prescriptions and recommendations for improving the regime classification approach. L467-480.

4. A discussion of the random forest emulator and suggestions for future users. L487-498.

**Some additional specific points:**

1) Abstract: "... that is purely data driven and requires few tuning parameters": I think this statement reflects more a limitation of the present study than a feature of the conceptual algorithm. There are significant tuning parameters (e.g., number of clusters, choice of predictors), but they are not explored in this study. I would argue that the method still requires substantial tuning of these parameters to be robust and useful compared to other methods, and some of the responses of the authors appear to agree with this. So I suggest to rephrase this statement to avoid giving the impression that this is a positive feature of the algorithm and that it can be run "out of the box".

We thank the reviewer for this comment. In order to resolve any confusion, and in light of the fact that we have highlighted several parts of the algorithm for future users to adjust and improve, we have removed any reference to 'few' tuning parameters.

2) Abstract, last sentence "... and it is shown to adequately capture the error features of the tracked wind.": I don't think it is clear what "adequate" means in this context, and I think it would be useful to be frank that the truth was available for training. I suggest to rephrase "..., and it is shown to capture some of the regime-dependent error features of the tracked wind in a setting where the truth is available for training the algorithm."

We thank the reviewer for this comment. We slightly disagree with the idea that the 'truth' was available for training. As discussed in the revised conclusions (L441-451), we trained the model on the first 1.5 months of the Nature Run data, and presented results applied on the last .5 months of the Nature Run; while this is a situation where we expect the training data to fairly accurately reflect the domain of the testing data, it is not in fact the same data and thus not what we would deem the 'truth'. Nevertheless, the term 'adequate' is inherently subjective and we have replaced this with 'provide accurate and useful error features of the tracked wind', which is shown in the analysis in section 4.

3) Reply to my earlier point 11: Thanks for providing the information on the significance of the standard deviation and bias statistics for the clusters included in the reply. This should be included in the revised manuscript, as it would help to substantiate the claim that the clusters identify statistically significantly different error behaviours.

Thank you for this comment. We have included our reply to point 11 in the paper at the end of section 4, L373-392 (as well as Figure 18)

List of relevant changes in manuscript (in order as they appear):

- 1. Abstract and Title:
  - a. Rewording of final sentence of abstract.
- 2. Section 4:
  - a. L: 376-396. Included a discussion on the statistical significance of the bias and standard errors values derived from the uncertainty model.
- 3. Conclusion and Discussion:
  - a. Re-written and greatly expanded conclusion section to include discussion of potential problem areas for those wishing to implement this methodology. Including, but not limited to, a discussion of:
    - i. A discussion of how humidity retrieval uncertainty would impact our model and approaches to address it. L417-431.
    - ii. A discussion of considerations future users must make when developing training data for the model. L432-466; L481-486.
    - iii. A list of specific prescriptions and recommendations for improving the regime classification approach. L467-480.
    - iv. A discussion of the random forest emulator and suggestions for future users. L487-498.
- 4. References:
  - a. Included reference to (Efron and Tibshirani, 1993).
- 5. Figures:
  - a. New Figure 18, which accompanies the new additions to Section 4.

[revised manuscript text omitted]

---

## Author Response (AR3)

"Using Machine Learning to Model Uncertainty for Water-Vapor Atmospheric Motion Vectors"

Teixeira et al.

Responses to Referee 2

We would like to thank the referee for the careful read of the paper and comments. Please see our responses below:

*I am, however, surprised by the paragraph L426-430 in the discussion, which lists only two main issues for "large scale applications" (I read this as "applications to real satellite-derived AMVs" - I suggest to rephrase this to make it clear what is meant): 1) uncertainties in the humidity field, and the 2) representativeness/variability of the training data. I would argue that the first and foremost problem is that the truth for both the wind and the humidity field is not available in applications with real data. In the present study the Nature Run serves as truth (by design), and hence the errors of the AMVs are completely known from the differences between AMVs and Nature-Run winds. These "true errors" are essential input to train the algorithm from the first 1.5 months of data. As soon as the Nature Run is replaced with real data (for the humidity field, but more importantly the wind field), the "true errors" are not available anymore, and other errors will be introduced (e.g., collocation and representativeness errors if other observations are used as proxy of the true wind, or analysis and representativeness errors if reanalysis data is used as proxy for the truth). As far as I can see, the algorithm has no knowledge that would allow it to separate the errors in the "reference" wind data from the errors in the AMVs alone. This aspect is related, but rather different from the one presently listed as 2nd issue. I think it should be mentioned separately here, and it may deserve some more explicit discussion. The following paragraphs are touching on this aspect (esp. L496-501), but the text offers little in terms of addressing this. It seems to me a fundamental question that is left unanswered in the paper, ie how to train the machine learning algorithm in the case of real-data applications, so that it is able to separate different sources of error in the training data.*

We thank the reviewer for their insightful comments and guidance throughout the review process. The question raised by the reviewer is indeed a fundamental one. It is our belief that, as with most machine learning approaches, a thorough understanding of the relative strengths and weaknesses of the training dataset is the most critical consideration for users. As the reviewer notes, this means not only ensuring that the training data is variable and diverse enough to encapsulate the entirety of the true domain, but possessing some understanding of how and where portions of the training dataset might be less representative of reality.

Assuming a user possesses the proper amount of domain knowledge, there are a few practical ways in which they could attempt to address this issue. One could, given the adequate resources and time, train the uncertainty model under various training datasets. While this would not necessarily give a user a greater understanding of the training data's relationship with the truth, the differences between the produced models would provide some quantification of the effect of the training data on the estimated uncertainties. If values were too divergent at similar points, that would indicate that the model is not particularly robust to errors in the training dataset and should be reconsidered. Similarly, if users have some quantified understanding of areas wherein the training dataset might be less useful (e.g., collocation errors), they could leverage this to inform the uncertainty model. For example, one could selectively subsample the dataset to underrepresent (or screen out) high error areas and so reduce their effect on the overall model. They could also include this information in the error model itself. Either way, such decisions would likely manifest themselves in the final uncertainty product.

None of these approaches, however, can fully resolve two inescapable truths of this machine-learning based approach to uncertainty modeling. First and foremost, as much as users should try to mitigate the potential for problems, there is always an underlying leap of faith that they have chosen a training dataset that adequately represents the truth in their application. Like any modeling approach, this methodology relies on a set of assumptions; this is one such assumption. This is why domain knowledge is critical in developing a similar uncertainty model. It is our belief that, under the right guidance and curation, such training data related issues would be outweighed by the improved contribution of the uncertainty estimates produced. Secondly, this approach produces an error characterization and not an error budget. This is an important distinction; we aim to produce an overall uncertainty of the AMV retrieval, not directly attribute that uncertainty (or components of it) to specific sources. While we have some physical understanding of sources of uncertainty in the AMVs, and this guides the development of the model, the algorithm itself is agnostic to these. As such, even if distinguishing between 'real AMV error' and 'training data error' were feasible in a practical sense (which, given the current practice in machine learning, is well beyond the scope of this work), the uncertainty model itself is not designed to discriminate between these.

A summarized version of this discussion has been added to the manuscript in lines 476-495.

List of relevant changes in manuscript (in order as they appear):
1. Conclusion and Discussion:
    a. Augmented paragraph (L476-495) detailing approaches and considerations for proxy-data error.

[revised manuscript text omitted]